# LipsFormer: Introducing Lipschitz Continuity to Vision Transformers

**Xianbiao Qi, Jianan Wang, Yihao Chen, Yukai Shi & Lei Zhang**[*]
International Digital Economy Academy (IDEA), Shenzhen, Guangdong, China.
`{qixianbiao,wangjianan,chenyihao,shiyukai,leizhang}@idea.edu.cn`

## Abstract

We present a Lipschitz continuous Transformer, called LipsFormer, to pursue training stability both theoretically and empirically for Transformer-based models. In contrast to previous practical tricks that address training instability by learning rate warmup, layer normalization, attention formulation, and weight initialization, we show that Lipschitz continuity is a more essential property to ensure training stability. In LipsFormer, we replace unstable Transformer component modules with Lipschitz continuous counterparts: CenterNorm instead of LayerNorm, spectral initialization instead of Xavier initialization, scaled cosine similarity attention instead of dot-product attention, and weighted residual shortcut. We prove that these introduced modules are Lipschitz continuous and derive an upper bound on the Lipschitz constant of LipsFormer. Our experiments show that LipsFormer allows stable training of deep Transformer architectures without the need of careful learning rate tuning such as warmup, yielding a faster convergence and better generalization. As a result, on the ImageNet 1K dataset, LipsFormer-Swin-Tiny based on Swin Transformer training for 300 epochs can obtain 82.7% without any learning rate warmup. Moreover, LipsFormer-CSwin-Tiny, based on CSwin, training for 300 epochs achieves a top-1 accuracy of 83.5% with 4.7G FLOPs and 24M parameters.

## 1 Introduction

Transformer [49] has been widely adopted in natural language processing (NLP) [6, 27, 40] for its great capability of capturing long-range dependencies with *self-attention*. Motivated by its success in NLP, Dosovitskiy *et al.* [17] introduced Vision Transformer (ViT) as a general backbone for computer vision tasks such as image classification [35, 53, 16], object detection [9, 59], and segmentation [12]. Nowadays, Transformer [49] remains the dominant architecture for NLP [5, 6, 40], computer vision [58, 35, 53, 16] and many other AI applications [42, 41, 31].

Despite its success, training Transformer remains challenging [33, 14] for practitioners: the training process can be prohibitively unstable, especially at the beginning of training. To address the root cause for training instability, we resort to examining Lipschitz continuity of Transformer components. Intuitively, a Lipschitz continuous network is finite in the rate of change and its Lipschitz constant is an useful indicator for training stability. As shown in [8, 7, 44], Lipschitz properties reveal intriguing behaviours of neural networks, such as robustness and generalization. In this work, we focus on the trainability issue of Transformer architectures by explicitly enforcing Lipschitz continuity at network initialization.

Previous works for overcoming Transformer training instability usually focus on one or a combination of its components which can be divided into four categories: (1) *improving normalization* [54, 33, 51]; Xiong *et al.* [54] has shown that, for a Transformer architecture, Pre-LayerNorm (Pre-LN) is more stable than Post-LayerNorm (Post-LN). Liu *et al.* [33] identified that Post-LN negatively influences training stability by amplifying parameter perturbations. They introduced adaptive model initialization (Admin) to mitigate the amplification effect. Likewise, Wang *et al.* [51] introduced DeepNorm and a depth-specific initialization to stabilize Post-LN. However, even with normalization improvements

---

[*]Corresponding author.

such as Admin and DeepNorm, learning rate warmup [20] is still a necessity to stabilize training. (2) *more stable attention* [28, 13]; Kim *et al.* [28] proved that the standard dot-product attention is not Lipschitz continuous and introduced an alternative L2 attention. (3) *re-weighted residual shortcut*; Bachlechner *et al.* [3] showed that a simple architecture change of gating each residual shortcut with a learnable zero-initialized parameter substantially stabilizes training. With ReZero, they were able to train extremely deep Transformers of 120 layers. (4) *careful weight initialization*; To avoid gradient exploding or vanishing at the beginning of training, Zhang *et al.* [60] proposed fixed-update initialization (Fixup) by rescaling a standard initialization. They also proved that Fixup could enable stable training of residual networks without normalization.

In this paper, we conduct a thorough analysis of Transformer architectures and propose a Lipschitz continuous Transformer called LipsFormer. In contrast to previous practical tricks that address training instability, we show that Lipschitz continuity is a more essential property to ensure training stability. We focus our investigation on the following Transformer components: LayerNorm, dot-product self-attention, residual shortcut, and weight initialization. For each analyzed module, we propose a Lipschitz continuous variant as a new building block for LipsFormer. The final LipsFormer network has an upper bound Lipschitz constant at initial stages of training. Such a Lipschitz guarantee has two implications: 1) we can train LipsFormer without using the common trick of learning rate warmup, yielding a faster convergence and better generalization; 2) Transformer is more unstable at the beginning of training. By ensuring initial network stability, we drastically increases the trainability of Transformer. Note that we could also enforce Lipschitz continuity during the whole training process by simply constraining updates on certain scaling parameters.

Our main contributions can be summarized as follows:

- We give a thorough analysis of key Transformer components: LayerNorm, self-attention, residual shortcut, and weight initialization. More importantly, we identify potential instability problems each module brings to the training difficulty and propose their Lipschitz continuous counterparts: CenterNorm, scaled cosine similarity attention, scaled residual shortcut, and spectral-based initialization. The proposed Lipschitz continuous modules can serve as drop-in replacements for a standard Transformer, such as Swin Transformer [35] and CSwin [16].

- We propose a Lipschitz continuous Transformer (LipsFormer) that can be stably trained without the need of carefully tuning the learning rate schedule. We derive theoretical Lipschitz constant upper bounds for both scaled cosine similarity attention and LipsFormer. The derivation provides a principled guidance for designing LipsFormer networks. We build LipsFormer-Swin and LipsFormer-CSwin based on Swin Transformer and CSwin individually.

- We validate the efficacy of the LipsFormer on ImageNet classification. We show empirically that LipsFormer can be trained smoothly without learning rate warmup. As a result, on the ImageNet-1K dataset, LipsFormer-Swin-Tiny training for 300 epochs can obtain a top-1 accuracy of 82.7% without any learning rate warmup. Moreover, LipsFormer-CSwin-Tiny training for 300 epochs achieves a top-1 accuracy of 83.5% with 4.7G FLOPs and 24M parameters.

## 2 PRELIMINARIES

In this section, we first define Lipschitz continuity and Lipschitz constant and then discuss several Lipschitz properties of a neural network. We use the denominator-layout notation throughout this paper. A sequence of $N$ elements is denoted as $\boldsymbol{X} = [\boldsymbol{x}_1; \ldots; \boldsymbol{x}_N]^\top \in \mathbb{R}^{N \times D}$, where each vector $\boldsymbol{x}_i \in \mathbb{R}^D, i \in \{1, ..., N\}$. Function transformation is parameterized by an associated weight matrix $\boldsymbol{W}$ and an affine transformation is denoted as $f(\boldsymbol{x}) = \boldsymbol{W}^\top \boldsymbol{x}$, where $\boldsymbol{W} \in \mathbb{R}^{D \times M}$.

**Definition 1.** *A function $f(\boldsymbol{x}, \boldsymbol{W}) : \mathbb{R}^D \to \mathbb{R}^M$ is Lipschitz continuous (L-Lipschitz) under a choice of p-norm $\| \cdot \|_p$ in the variable $\boldsymbol{x}$ if there exists a constant $L$ such that for all $(\boldsymbol{x}_1, \boldsymbol{W})$ and $(\boldsymbol{x}_2, \boldsymbol{W})$ in the domain of $f$,*

$$\|f(\boldsymbol{x}_1, \boldsymbol{W}) - f(\boldsymbol{x}_2, \boldsymbol{W})\|_p \leq L \|\boldsymbol{x}_1 - \boldsymbol{x}_2\|_p,$$

where the smallest value of $L$ that satisfies the inequality is called the Lipschitz constant of $f$. To emphasize that the Lipschitz constant with respect to $\boldsymbol{x}$ depends on $\boldsymbol{W}$ and the choice of $p$, we denote $L$ as $\mathrm{Lip}_p(f_{\boldsymbol{x}}(\boldsymbol{W}))$. A function is generally referred to as expansive, non-expansive, and contractive in the variable $\boldsymbol{x}$ for $\mathrm{Lip}_p(f_{\boldsymbol{x}}(\boldsymbol{W})) > 1$, $\mathrm{Lip}_p(f_{\boldsymbol{x}}(\boldsymbol{W})) \leq 1$, and $\mathrm{Lip}_p(f_{\boldsymbol{x}}(\boldsymbol{W})) < 1$, respectively,

exhibiting characteristic differences in the change rate of its output. Contemporary neural networks are rarely Lipschitz continuous under the influence of any constituent non-Lipschitz module. Even if a network is Lipschitz continuous, calculating its Lipschitz constant exactly is a challenging task [50].

According to Definition 1, the Lipschitz constant of $f(\boldsymbol{x}, \boldsymbol{W})$ with respect to $\boldsymbol{x}$ can be expressed as,

$$\mathrm{Lip}_p(f_{\boldsymbol{x}}(\boldsymbol{W})) = \sup_{\boldsymbol{x}_1 \neq \boldsymbol{x}_2 \in \mathbb{R}^D} \frac{\|f(\boldsymbol{x}_1, \boldsymbol{W}) - f(\boldsymbol{x}_2, \boldsymbol{W})\|_p}{\|\boldsymbol{x}_1 - \boldsymbol{x}_2\|_p}.$$

Exact computation of the above equation is an NP-hard problem. For subsequent analyses, We use $p = 2$ by default unless specified and suppress $p$ to reduce clutter, but our conclusion can be easily extended to other choices of $p$.

**Lemma 1.** *Given $\boldsymbol{W}$, let $f(\boldsymbol{x}, \boldsymbol{W}) : \mathbb{R}^D \to \mathbb{R}$ be a continuously differentiable function and $\mathrm{Lip}(f_{\boldsymbol{x}}(\boldsymbol{W}))$ be its Lipschitz constant with respect to $\boldsymbol{x}$. According to the mean value theorem, we have the following inequality,*

$$\|\nabla_{\boldsymbol{x}} f(\boldsymbol{x}, \boldsymbol{W})\| \leq \mathrm{Lip}(f_{\boldsymbol{x}}(\boldsymbol{W})), \forall \boldsymbol{x} \in \mathbb{R}^D,$$

where $\|\nabla_{\boldsymbol{x}} f(\boldsymbol{x}, \boldsymbol{W})\|$ is the gradient norm of $f(\boldsymbol{x}, \boldsymbol{W})$ with respect to $\boldsymbol{x}$.

From Lemma 1, we can see that a practical method to compute the Lipschitz constant of a continuously differentiable function is to compute its maximum gradient norm. To prove a function is not Lipschitz, it is sufficient to show that its gradient norm is not bounded. For example, $f(x) = \frac{1}{x}$ and $f(x) = x^2$ are not Lipschitz continuous for $x \in (0, \infty)$, because their gradient can be arbitrarily large as $x$ approaches 0 and $\infty$, respectively.

**Definition 2.** *Let $F(\boldsymbol{x}, \{\boldsymbol{W}^l, l = 1, \dots, L\}) : \mathbb{R}^D \to \mathbb{R}$ be an L-layer neural network defined as a composite function with L transformation functions:*

$$F(\boldsymbol{x}, \{\boldsymbol{W}^l, l = 1, \dots, L\}) = f^L \left( f^{L-1} \left( \dots f^1 \left( \boldsymbol{x}, \boldsymbol{W}^1 \right), \boldsymbol{W}^2 \right) \dots, \boldsymbol{W}^L \right),$$

where $\{\boldsymbol{W}^l, l = 1, \dots, L\}$ is the parameter set, and $f^l$ is the transformation function of the $l$-th layer.

For an affine transformation $f(\boldsymbol{x}, \boldsymbol{W}) = \boldsymbol{W}^\top \boldsymbol{x}$, its Lipschitz constant is,

$$\mathrm{Lip}_p(f_{\boldsymbol{x}}(\boldsymbol{W})) = \sup_{\|\boldsymbol{x}\|_p = 1} \|\boldsymbol{W}^\top \boldsymbol{x}\|_p = \left\{ \begin{array}{ll} \sigma_{\max}(\boldsymbol{W}), & \text{if } p = 2 \\ \max_i \sum_j |\boldsymbol{W}_{ij}| & \text{if } p = \infty \end{array} \right. \tag{1}$$

where $\sigma_{\max}(\boldsymbol{W})$ is the largest eigenvalue of $\boldsymbol{W}$.

Many common activation functions such as Sigmoid, Tanh, ReLU, and GELU are 1-Lipschitz. See Appendix A.1 for a simple illustration.

**Lemma 2.** *Given the Lipschitz constant of each transformation function in a network F, the following inequality holds*

$$\mathrm{Lip}(F_{\boldsymbol{x}}(\{\boldsymbol{W}^l, l = 1, \dots, L\})) \leq \prod_{l=1}^{L} \mathrm{Lip}(f_{\boldsymbol{x}}^l(\boldsymbol{W}^l)).$$

From Lemma 2, the Lipschitz constant of a network is upper bounded by the product of each layer's Lipschitz constant. This multiplicative nature gives us an insight into understanding why deeper networks usually suffer more severe training instability: if a network's constituent layers are expansive, the upper bound of its Lipschitz constant increases monotonically with its network depth. We refer the interested readers to [18, 30] for estimating tighter bounds of deep neural networks.

## 3 AN ASSUMPTION FOR TRAINING STABILITY

Our design philosophy for LipsFormer is based on the following assumption.

**Assumption 1.** *A network should satisfy the following Lipschitz conditions for training stability,*

1. $\quad \|f(\boldsymbol{x}_1, \boldsymbol{W}) - f(\boldsymbol{x}_2, \boldsymbol{W})\| \leq \mathrm{Lip}(f_{\boldsymbol{x}}(\boldsymbol{W}))\|\boldsymbol{x}_1 - \boldsymbol{x}_2\|,$

2. $\quad \|f(\boldsymbol{x}, \boldsymbol{W}_1) - f(\boldsymbol{x}, \boldsymbol{W}_2)\| \leq \mathrm{Lip}(f_{\boldsymbol{W}}(\boldsymbol{x}))\|\boldsymbol{W}_1 - \boldsymbol{W}_2\|.$

The first inequality focuses on the forward process and assumes that a stable network should satisfy Lipschitz continuity with respect to its input $\boldsymbol{x}$: a small perturbation of its input should not lead to a drastic change of its output. Guaranteeing smoothness is vital for guarding a network's generalization ability.

For the second inequality, recall that the forward process of a typical neural network propagates computation as $\boldsymbol{x}^{l+1} = (\boldsymbol{W}^{l+1})^{\top} \boldsymbol{x}^l$, where $\boldsymbol{x}^l$ and $\boldsymbol{W}^{l+1}$ are the input and weight matrix of Layer $l+1$. Since common non-linearities are 1-Lipschitz, we drop non-linear activations here for simplicity. To backpropagate the network loss $\mathcal{L}$, we have

$$\frac{\partial \mathcal{L}}{\partial \boldsymbol{x}^l} = \boldsymbol{W}^{l+1} \frac{\partial \mathcal{L}}{\partial \boldsymbol{x}^{l+1}}, \quad \frac{\partial \mathcal{L}}{\partial \boldsymbol{W}^{l+1}} = \boldsymbol{x}^l \left(\frac{\partial \mathcal{L}}{\partial \boldsymbol{x}^{l+1}}\right)^{\top}.$$

Gradient descent updates network weights according to $\boldsymbol{W} \leftarrow \boldsymbol{W} - lr \times \frac{\partial \mathcal{L}}{\partial \boldsymbol{W}}$. As demonstrated above, any value explosion will propagate with the chain derivation: if $\frac{\partial \mathcal{L}}{\partial \boldsymbol{x}^{l+1}}$ is unbounded, $\frac{\partial \mathcal{L}}{\partial \boldsymbol{x}^l}$ and $\frac{\partial \mathcal{L}}{\partial \boldsymbol{W}^{l+1}}$ will consequently be unbounded. Meanwhile, if $\frac{\partial \mathcal{L}}{\partial \boldsymbol{W}^{l+1}}$ is not bound, it will largely influence the back-propagation chain in the next iteration. This justifies the second inequality for the purpose of training stability.

Intuitively, guaranteeing that a network's output does not change too much under small perturbations of either its input or network weights induces a more stable training process. In this work, we focus on satisfying the first inequality in Assumption 1 for Transformer architectures.

## 4 LIPSFORMER

A Lipschitz continuous Transformer (LipsFormer) requires all of its constituent modules to be Lipschitz continuous according to Lemma 2. In this section, we analyze key Transformer components and introduce their Lipschitz continuous counterparts when any Lipschitz continuity is violated.

### 4.1 LIPSCHITZ CONTINUOUS MODULES

#### 4.1.1 CENTERNORM INSTEAD OF LAYERNORM

LayerNorm [2] is the most widely used normalization method in Transformer. It is defined as

$$\mathrm{LN}(\boldsymbol{x}) = \boldsymbol{\gamma} \odot \boldsymbol{z} + \boldsymbol{\beta}, \text{ where } \boldsymbol{z} = \frac{\boldsymbol{y}}{\mathrm{Std}(\boldsymbol{y})} \text{ and } \boldsymbol{y} = \left(\boldsymbol{I} - \frac{1}{D}\mathbf{1}\mathbf{1}^{\top}\right)\boldsymbol{x},$$

where $\boldsymbol{x}, \boldsymbol{y} \in \mathbb{R}^D$, $\mathrm{Std}(\boldsymbol{y})$ is the standard deviation of the mean-subtracted input $\boldsymbol{y}$, and $\odot$ is an element-wise product. $\boldsymbol{\gamma}$ and $\boldsymbol{\beta}$ are initialized to $\mathbf{1}$ and $\mathbf{0}$ respectively. For simplicity, we drop $\boldsymbol{\gamma}$ and $\boldsymbol{\beta}$ from analysis because they can be explicitly constrained within any pre-defined range.

By taking partial derivatives, the Jacobian matrix of $\boldsymbol{z}$ with respect to $\boldsymbol{x}$ is,

$$\boldsymbol{J}_{\boldsymbol{z}}(\boldsymbol{x}) = \frac{\partial \boldsymbol{z}}{\partial \boldsymbol{x}} = \frac{\partial \boldsymbol{z}}{\partial \boldsymbol{y}}\frac{\partial \boldsymbol{y}}{\partial \boldsymbol{x}} = \frac{1}{\mathrm{Std}(\boldsymbol{y})}\left(\boldsymbol{I} - \frac{1}{D}\mathbf{1}\mathbf{1}^{\top}\right)\left(\boldsymbol{I} - \frac{\boldsymbol{y}\boldsymbol{y}^{\top}}{\|\boldsymbol{y}\|_2^2}\right).$$

The equation above shows that LayerNorm is not Lipschitz continuous because when $\mathrm{Std}(\boldsymbol{y})$ approaches 0, the values in the Jacobian matrix will approach $\infty$, causing severe training instability. On the other end, when $\mathrm{Std}(\boldsymbol{y})$ is very large, training will be hindered by LayerNorm as gradients become extremely small. Also note that backpropagating through LayerNorm is slow due to poor parallelization when computing the Jacobian matrix, especially for the term $I - \frac{\boldsymbol{y}\boldsymbol{y}^{\top}}{\|\boldsymbol{y}\|_2^2}$.

In practice, we notice that a single LayerNorm operation could cause severe training instability without learning rate warmup. The underlying reason is that LayerNorm is not Lipschitz continuous and some ill-defined input with zero variance will lead to a Jacobian matrix filled with infinity. To stabilize training by enforcing Lipschitz continuity, we introduce CenterNorm as,

$$\mathrm{CN}(\boldsymbol{x}) = \boldsymbol{\gamma} \odot \frac{D}{D-1}\left(\boldsymbol{I} - \frac{1}{D}\mathbf{1}\mathbf{1}^{\top}\right)\boldsymbol{x} + \boldsymbol{\beta}, \tag{2}$$

where $D$ is the dimension of $\boldsymbol{x}$. The Jacobian matrix $\frac{\partial \operatorname{CN}(\boldsymbol{x})}{\partial \boldsymbol{x}}$ contains a term $\frac{D}{D-1}\left(\boldsymbol{I} - \frac{1}{D}\mathbf{1}\mathbf{1}^\top\right)$ where $\frac{D}{D-1}$ is a heuristic to avoid the eigenvalue contraction from $\left(\boldsymbol{I} - \frac{1}{D}\mathbf{1}\mathbf{1}^\top\right)$. It is easy to verify that,

$$\|\operatorname{CN}(\boldsymbol{x}_1) - \operatorname{CN}(\boldsymbol{x}_2)\| \leq \operatorname{Lip}(\operatorname{CN}_{\boldsymbol{x}})\|\boldsymbol{x}_1 - \boldsymbol{x}_2\|,$$

where $\operatorname{Lip}(\operatorname{CN}_{\boldsymbol{x}}) = \frac{D}{D-1}$ for $\boldsymbol{\gamma} = \mathbf{1}$ and $\boldsymbol{\beta} = \mathbf{0}$. As most deep neural networks are dealing with high dimensional data with $D \gg 1$, we make a simplification that $\operatorname{Lip}(\operatorname{CN}_{\boldsymbol{x}})$ is 1-Lipschitz for later discussions. CenterNorm is by design Lipschitz continuous at initialization. To guarantee its Lipschitz continuity through training we could simply constrain $\boldsymbol{\gamma}$ and $\boldsymbol{\beta}$ to a pre-defined range.

### 4.1.2 Scaled Cosine Similarity Attention

Self-attention [49] is a key component of Transformer, helping capture long-range relationships within data. In practice, people use multi-head attention to effectively capture such relationships under different contexts. Since multi-head attention is a linear combination of multiple single-head attention outputs, for simplicity, we focus our analysis on single-head attention, which is defined as,

$$\operatorname{Attn}(\boldsymbol{X}, \boldsymbol{W}^Q, \boldsymbol{W}^K, \boldsymbol{W}^V) = \operatorname{softmax}\left(\frac{\boldsymbol{X}\boldsymbol{W}^Q\left(\boldsymbol{X}\boldsymbol{W}^K\right)^\top}{\sqrt{D}}\right)\boldsymbol{X}\boldsymbol{W}^V, \tag{3}$$

where $\boldsymbol{W}^Q, \boldsymbol{W}^K, \boldsymbol{W}^V$ are the projection matrices to transform $\boldsymbol{X}$ into query, key, and value matrices, respectively. Intuitively, every token aggregates information from all the visible tokens by computing a weighted sum of the values of the visible tokens according to the similarity between its query and each visible token's key. The similarity between the $i$-th query $\mathbf{q}_i$ and $j$-th key $\mathbf{k}_j$ is denoted as $\boldsymbol{P}_{ij} \propto \boldsymbol{x}_i^\top \boldsymbol{W}^Q(\boldsymbol{W}^K)^\top \boldsymbol{x}_j$.

In [28], Kim *et al.* proved that the standard dot-product self-attention is *not* Lipschitz continuous and introduced an alternative L2 self-attention that is Lipschitz continuous. Here we use a scaled cosine similarity attention, which is defined as,

$$\operatorname{SCSA}(\boldsymbol{X}, \boldsymbol{W}^Q, \boldsymbol{W}^K, \boldsymbol{W}^V, \nu, \tau) = \nu\boldsymbol{P}\boldsymbol{V}, \text{ where } \boldsymbol{P} = \operatorname{softmax}\left(\tau\boldsymbol{Q}\boldsymbol{K}^\top\right),$$

$$\boldsymbol{Q} = \begin{bmatrix} - & \mathbf{q}_1^\top & - \\ & \vdots & \\ - & \mathbf{q}_N^\top & - \end{bmatrix} \quad \boldsymbol{K} = \begin{bmatrix} - & \mathbf{k}_1^\top & - \\ & \vdots & \\ - & \mathbf{k}_N^\top & - \end{bmatrix} \quad \boldsymbol{V} = \begin{bmatrix} - & \mathbf{v}_1^\top & - \\ & \vdots & \\ - & \mathbf{v}_N^\top & - \end{bmatrix},$$

where $\nu$ and $\tau$ are predefined or learnable scalars; $\boldsymbol{Q}, \boldsymbol{K}, \boldsymbol{V}$ are $\ell^2$ row-normalized: $\mathbf{q}_i, \mathbf{k}_i, \mathbf{v}_i = \frac{(\mathbf{x}_i^\top \boldsymbol{W}^Q)^\top}{\sqrt{\|\mathbf{x}_i^\top \boldsymbol{W}^Q\|^2 + \epsilon}}, \frac{(\mathbf{x}_i^\top \boldsymbol{W}^K)^\top}{\sqrt{\|\mathbf{x}_i^\top \boldsymbol{W}^K\|^2 + \epsilon}}, \frac{(\mathbf{x}_i^\top \boldsymbol{W}^V)^\top}{\sqrt{\|\mathbf{x}_i^\top \boldsymbol{W}^V\|^2 + \epsilon}}; \epsilon$ is a smoothing factor to guarantee the validity of cosine similarity computation even when $\|\mathbf{x}_i^\top \boldsymbol{W}^Q\| = 0$. For arbitrary pair of rows of $\boldsymbol{Q}$ and $\boldsymbol{K}$ denoted as $\mathbf{q}_i$ and $\mathbf{k}_j$, the cosine similarity on their $\ell^2$-normalized vectors is proportional to their L2 dot product. The upper bound of SCSA's Lipschitz constant with respect to $\|\cdot\|_2$ and $\|\cdot\|_\infty$ is the following,

**Theorem 1.** *Single-head scaled cosine similarity attention is Lipschitz continuous, its* $\operatorname{Lip}_\infty$ *and* $\operatorname{Lip}_2$ *are upper bounded by the following inequalities,*

$$\operatorname{Lip}(SCSA)_\infty \leq N^2\sqrt{D}\nu\tau\epsilon^{-\frac{1}{2}}\|\boldsymbol{W}^K\|_\infty + N\sqrt{D}\nu\tau\epsilon^{-\frac{1}{2}}\|\boldsymbol{W}^Q\|_\infty + 2N\nu\epsilon^{-\frac{1}{2}}\|\boldsymbol{W}^{V^\top}\|_\infty,$$

$$\operatorname{Lip}(SCSA)_2 \leq 2N(N-1)\nu\tau\epsilon^{-\frac{1}{2}}\|\boldsymbol{W}^K\|_2 + 2(N-1)\nu\tau\epsilon^{-\frac{1}{2}}\|\boldsymbol{W}^Q\|_2 + 2N\nu\epsilon^{-\frac{1}{2}}\|\boldsymbol{W}^{V^\top}\|_2.$$

Proof of Theorem 1 can be found in Appendix H. For multi-head attention, we heuristically scale head feature concatenation by $\frac{1}{K}$ where $K$ is the number of heads. Please refer to Appendix A.2 for more details.

### 4.1.3 Weighted Residual Shortcut

Residual block [24] is a common component of contemporary neural networks [35, 53, 49]. It has been proven effective in avoiding gradient vanishing, especially when training deep networks. A standard residual shortcut block is defined as,

$$\operatorname{RS}(\boldsymbol{x}, \boldsymbol{W}) = \boldsymbol{x} + f(\boldsymbol{x}, \boldsymbol{W}).$$

The Lipschitz constant of a residual shortcut block with respect to $\boldsymbol{x}$ is $\text{Lip}(RS_{\boldsymbol{x}}(\boldsymbol{W})) = 1 + \text{Lip}(f_{\boldsymbol{x}}(\boldsymbol{W}))$. For any non-degenerate Lipschitz continuous function $f(\boldsymbol{x}, \boldsymbol{W})$, its Lipschitz constant is greater than 0, hence a residual block is strictly expansive. According to Lemma 2, stacking $L$ identical residual blocks alone will grow the upper bound of a network's Lipschitz constant exponentially to $\text{Lip}(RS_{\boldsymbol{x}}(\boldsymbol{W}))^L$, causing substantial vulnerability to forward value explosion. One way to mitigate such an instability is to constraint the Lipschitz constant of the residual path to be much smaller than 1, especially at the beginning of training when the network is undergoing fast changes via learning. In this paper, we explicitly multiply the residual path with a scale factor initialized to a small value such as 0.1 and 0.2. We define the weighted residual shortcut as,

$$\text{WRS}(\boldsymbol{x}, \boldsymbol{W}) = \boldsymbol{x} + \boldsymbol{\alpha} \odot f(\boldsymbol{x}, \boldsymbol{W}), \tag{4}$$

where $\boldsymbol{\alpha}$ is a learnable parameter vector with the same dimension as the channel size of $\boldsymbol{x}$.

It is easy to verify that

$$\| \text{WRS}(\boldsymbol{x}_1, \boldsymbol{W}) - \text{WRS}(\boldsymbol{x}_2, \boldsymbol{W}) \| \leq \text{Lip}(\text{WRS}_{\boldsymbol{x}}(\boldsymbol{W})) \|\boldsymbol{x}_1 - \boldsymbol{x}_2\|,$$

where $\text{Lip}(\text{WRS}_{\boldsymbol{x}}(\boldsymbol{W})) = 1 + \max(\boldsymbol{\alpha})$ when $\text{Lip}(f_{\boldsymbol{x}}(\boldsymbol{W})) = 1$.

As training progresses, $\boldsymbol{\alpha}$ changes as part of the learning process. We could easily constrain $\boldsymbol{\alpha}$ to a pre-defined range to ensure the Lipschitz continuity of a network during the whole training process. Note that re-weighting shortcut and residual path has been explored before: in [51, 33], the authors redefine a residual block as $\boldsymbol{\alpha} \odot \boldsymbol{x} + f(\boldsymbol{x}, \boldsymbol{W})$ to alleviate the LayerNorm instability; ReZero [3] uses a similar formulation as Equation 4 to speed up convergence, where $\alpha$ is a scalar instead of a vector. Our formulation is motivated by decreasing the Lipschitz constant of a network, instead of being a practical trick. It provides a more principled guidance to network design. For example, when training a very deep network, a smaller $\boldsymbol{\alpha}$ would be justified for the purpose of training stabilization.

### 4.1.4 SPECTRAL INITIALIZATION FOR CONVOLUTION AND FEED-FORWARD CONNECTION

Both convolution and feed-forward connection are compositions of affine transformations. As shown in Equation 1, affine transformation is Lipschitz continuous, hence by Lemma 2 both convolution and feed-forward connection are Lipschitz continuous.

Note that a careful initialization is important for successfully training a neural network. Many initialization methods have been proposed before such as Xavier [19] and Kaiming [23] initialization. Inspired by spectral norm regularization [56], we introduce a 1-Lipschitz initialization called spectral initialization,

$$\boldsymbol{W}_{si} = \frac{\boldsymbol{W}}{\sigma_{max}(\boldsymbol{W})}, \tag{5}$$

where $\boldsymbol{W}$ is Xavier-norm initialized and $\sigma_{max}(\boldsymbol{W})$ is its largest eigenvalue. For affine transformation $f(\boldsymbol{x}, \boldsymbol{W}_{si}) = \boldsymbol{W}_{si}^\top \boldsymbol{x}$, its Lipschitz constant satisfies the following inequality,

$$\| \boldsymbol{W}_{si}^\top \boldsymbol{x}_1 - \boldsymbol{W}_{si}^\top \boldsymbol{x}_2 \| \leq \text{Lip}(f_{\boldsymbol{x}}(\boldsymbol{W}_{si})) \|\boldsymbol{x}_1 - \boldsymbol{x}_2\|,$$

where $\text{Lip}(f_{\boldsymbol{x}}(\boldsymbol{W}_{si})) = 1$ at initialization. We use spectral initialization on all convolutions and feed-forward connections.

### 4.2 LIPSFORMER

#### 4.2.1 LIPSFORMER BLOCK

We start by introducing the main building block for LipsFormer. As shown in Figure 1, each LipsFormer block (LipsBlock) is composed of three sub-modules: convolution blocks (lightweight depth-wise and element-wise convolution), scaled cosine similarity attention, and feed-forward connection. CenterNorm operator is optional after each sub-module. In this work we apply CenterNorm after scaled cosine similarity attention and feed-forward connection. Within each residual block, residual path is re-weighted with a learnable $\alpha$ and randomly dropped with probability $p$ during training as indicated by dashed lines. For the convolution blocks, we use a $7 \times 7$ depth-wise and a $1 \times 1$ element-wise convolution. Ablation study on each component can be found in Sec. 5.3.

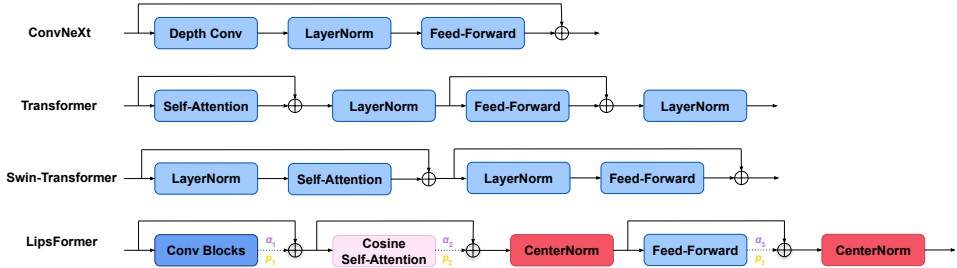

FIGURE 1: Comparison of a LipsFormer block with ConvNeXt, Transformer and Swin-Transformer Blocks. We use different colors to mark our Lipschitz improvements.

| | |
|---|---|
| Post-Norm | $\boldsymbol{x}_{i+1} = \text{LayerNorm}(\boldsymbol{x}_i + f(\boldsymbol{x}_i))$ |
| Pre-Norm | $\boldsymbol{x}_{i+1} = \boldsymbol{x}_i + f(\text{LayerNorm}(\boldsymbol{x}_i))$ |
| **LipsFormer** | $\boldsymbol{x}_{i+1} = \text{CenterNorm}(\boldsymbol{x}_i + \text{DropPath}_{p_i}(\boldsymbol{\alpha}_i f(\boldsymbol{x}_i)))$ |

TABLE 1: Various forms of residual blocks for Transformer architectures. As illustrated in Figure 1, $f$ represents a transformation function $\in$ {self-attention, feed-forward}. For LipsFormer, $f \in$ {scaled cosine similarity attention, feed-forward, convolution blocks}.

In Table 1 we compare the LipsFormer residual block with commonly used Post-Norm and Pre-Norm residual blocks. CenterNorm and scaled cosine similarity attention are Lipschitz continuous counterparts for LayerNorm and dot-product attention. Weighted residual connection and DropPath [29] are used to constrain the Lipschitz constant of a deep LipsFormer network.

### 4.2.2 OVERALL ARCHITECTURE OF LIPSFORMER

In general, LipsFormer follows the architecture of Swin Transformer v1. We start by processing an input image with non-overlapped convolutional token embedding ($4 \times 4$ convolution with stride 4) to obtain a feature representation with resolution $\frac{H}{4} \times \frac{W}{4}$. Then the main computation passes four stages where each stage consists of a pre-defined number of LipsFormer blocks as shown in Figure 1. Between consecutive stages, we reduce the output resolution by 2 and double the size of output channels by a $2 \times 2$ non-overlapped convolution with stride 2.

We build three variants of LipsFormer in correspondence with CSwin Transformer [16] as detailed in Appendix Table 4: LipsFormer-CSwin-Tiny (LipsFormer-CSwin-T) , LipsFormer-CSwin-Small (LipsFormer-CSwin-S), and LipsFormer-CSwin-Base (LipsFormer-CSwin-B). The number of Lips-Former blocks within the four computation stages are [1, 2, 21, 1] for LipsFormer-CSwin-T, [2, 4, 32, 2] for LipsFormer-CSwin-S and LipsFormer-CSwin-B. The overall architecture of LipsFormer is illustrated in Figure 3 of Appendix B. We can also build LipsFormer on Swin Transformer, more experiments about LipsFormer-Swin can be found in Appendix.

### 4.2.3 LIPSCHITZ CONSTANT OF LIPSFORMER

As illustrated in Figure 3, LipsFormer includes four computation stages, each starting with patch merging followed by a pre-defined number of LipsFormer blocks. Feed-forward connection, convolution, and patch merging are Lipschitz continuous operators. With spectral initialization these affine transformations are 1-Lipschitz at the beginning of training, hence are dropped from analysis. For the Lipschitz constant of the LipsFormer, we have the following theorem.

**Theorem 2.** *For a LipsFormer with $S$ stages where the $s$-th stage has $M_s$ residual blocks, when $\alpha$ is set to $\frac{1}{\sum_{s=1}^{S} M_s}$, the Lipschitz constant of the LipsFormer is upper bounded by $\exp(\kappa)$, where $\kappa = \max(\{\text{Lip}(f_i) : i = 1, \ldots, \sum_{s=1}^{S} M_s\})$.*

Proof of Theorem 2 is in Appendix I. Theorem 2 suggests that 1) Deeper networks with more residual blocks should initialize with a smaller $\alpha$ to avoid exponential growth of its Lipschitz constant; 2) To control the Lipschitz constant of Lipsformer we should focus on constraining the Lipschitz constant $\text{Lip}(f_i)$ of each constituent layer, especially the one with the largest Lipschitz constant.

## 5 EXPERIMENTS

### 5.1 DATASET AND TRAINING SETUP

We evaluate LipsFormer-CSwin on the standard ImageNet-1K [15] dataset, which consists of 1.28M images and 1,000 classes. We adopt a similar training strategy as in CSwin Transformer [16] for a fair comparison. Specifically, we use the AdamW [38] optimizer with weight decay 0.05 for LipsFormer-CSwin-T/S and 0.1 for LipsFormer-CSwin-B. By default, all our models are trained for 300 epochs with an input image size of $224 \times 224$. For LipsFormer-CSwin, the training batch size is 2048 and the initial learning rate is 0.002 with a standard cosine learning rate decay [37] *without* learning rate warmup [37]. We apply stochastic depth [26] for LipsFormer-CSwin-T, LipsFormer-CSwin-S, and LipsFormer-CSwin-B, with a maximum DropPath rate of 0.2, 0.4, and 0.5, respectively. For ablation study, we train each model for 100 epochs for efficiency. See Appendix C for more details.

### 5.2 COMPARISON WITH STATE-OF-THE-ART MODELS

Table 2 reports the LipsFormer-CSwin results compared with state-of-the-art CNN and Transformer models. We evaluate all three variants of LipsFormer-CSwin against state-of-the-art models of similar sizes: Tiny ($< 32M$ parameters), Small (31-64M parameters), and Base (56-96M parameters).

| Method | Param. | FLOPs | Top-1 | Method | Param. | FLOPs | Top-1 | Method | Param. | FLOPs | Top-1 |
|---|---|---|---|---|---|---|---|---|---|---|---|
| RegNetY-4G [39] | 21M | 4.0G | 80.0 | RegNetY-8G [39] | 39M | 8.0G | 81.7 | RegNetY-16G [39] | 84M | 16.0G | 82.9 |
| EffNet-B4 [45] | 19M | 4.2G | 82.9 | EffNet-B5 [45] | 30M | 9.9G | 83.6 | EffNet-B7 [45] | 66M | 37.0G | 84.3 |
| ConvNeXt-T [36] | 28M | 4.5G | 82.1 | ConvNeXt-S [36] | 50M | 8.7G | 83.1 | ConvNeXt-B [36] | 89M | 15.4G | 83.8 |
| SE-CoTNetD-50 [32] | 23M | 4.1G | 81.6 | SE-CoTNetD-101 [32] | 41M | 8.5G | 83.2 | SE-CoTNetD-152 [32] | 56M | 17.0G | 84.0 |
| DeiT-S [46] | 22M | 4.6G | 79.8 | - | - | - | - | DeiT-B [46] | 87M | 17.5G | 81.8 |
| T2T-14 [57] | 24M | 5.2G | 81.5 | T2T-19 [57] | 39M | 8.9G | 81.9 | T2T-24 [57] | 64M | 14.1G | 82.3 |
| TNT-T [21] | 24M | 5.2G | 81.3 | - | - | - | - | TNT-B [21] | 66M | 14.1G | 82.8 |
| DeepViT [62] | 27M | 6.2G | 82.3 | DeepViT [62] | 55M | 12.5G | 83.1 | | | | |
| Swin-T [35] | 29M | 4.5G | 81.3 | Swin-S [35] | 50M | 8.7G | 83.0 | Swin-B [35] | 88M | 15.4G | 83.5 |
| CvT-13 [53] | 20M | 4.5G | 81.6 | CvT-21 [53] | 32M | 7.1G | 82.5 | - | - | - | - |
| NesT-T [61] | 17M | 5.8G | 81.5 | NesT-S [61] | 38M | 10.4G | 83.3 | NesT-B [61] | 68M | 17.9G | 83.8 |
| XCiT-S12 [1] | 26M | 4.8G | 82.0 | XCiT-S24 [1] | 48M | 9.1G | 82.6 | XCiT-M24 [1] | 84M | 16.2G | 82.7 |
| CrossViT-15 [11] | 27M | 5.8G | 81.5 | CrossViT-18 [11] | 44M | 9.0G | 82.8 | - | - | - | - |
| RegionViT-T [10] | 14.3M | 2.7G | 81.5 | RegionViT-S [10] | 30.6M | 5.3G | 82.6 | RegionViT-B [10] | 72.7M | 13.0G | 83.2 |
| Focal-T [55] | 29M | 4.9G | 82.2 | Focal-S [55] | 51M | 9.1G | 83.5 | Focal-B [55] | 90M | 16.0G | 83.8 |
| CSwin-T [16] | 23M | 4.3G | 82.7 | CSwin-S [16] | 35M | 6.9G | 83.6 | CSwin-B [16] | 78M | 15.0G | 84.2 |
| CrossFormer-S [52] | 31M | 4.9G | 82.5 | CrossFormer-B [52] | 52M | 9.2G | 83.4 | CrossFormer-L [52] | 92M | 16.1G | 84.0 |
| ViT-S (DeiT III) [47] | 22M | 4.6G | 81.4 | - | - | - | - | ViT-B (DeiT III) [47] | 87M | 17.5G | 83.8 |
| NAT-T [22] | 28M | 4.3G | 83.2 | NAT-S [22] | 51M | 7.8G | 83.7 | NAT-B [22] | 90M | 13.7G | 84.3 |
| LipsFormer-CSwin-T (ours) | 24M | 4.7G | 83.5 | LipsFormer-CSwin-S (ours) | 38M | 7.6G | 83.8 | LipsFormer-CSwin-B | 83M | 16.3G | 84.6 |

TABLE 2: Comparison of different models with input resolution $224^2$ on ImageNet-1K classification. Red indicates the best result and blue indicates the second best result.

Compared with previous state-of-the-art Vision Transformer models, LipsFormer-CSwin attains a higher classification accuracy on all its model variants. For instance, LipsFormer-CSwin-T obtains a 83.5% Top-1 accuracy that outperforms CSwin-T by 0.8%, ViT-S by 2.1% and NAT-T by 0.3%. LipsFormer-CSwin-T also outperforms recently improved CNN architectures, such as ConvNeXt-T and EffNet-B4 by 1.4% and 0.6%, respectively. LipsFormer-CSwin-T has fewer parameters than NAT-T and ConvNeXt-T. LipsFormer-CSwin-B also outperforms its counterparts, including Swin-B, CrossFormer-L, ConvNeXt-B, DeiT-B, ViT-B, and NAT-B with fewer parameters. Also note that all the other Transformer models use learning rate warmup, but LipsFormer-CSwin does not.

### 5.3 ABLATION STUDY

We conduct extensive ablation study on each key component of LipsFormer-CSwin as shown in Table 3. We use LipsFormer-CSwin-T for ablation study and all results in this comparison are trained for 100 epochs without learning rate warmup, except for ablation on warmup.

**Warmup.** In previous experiments we do *not* use learning rate warmup when training LipsFormer-CSwin. Theoretically, warmup is not needed given LipsFormer's appealing stabilization guarantee. According to the results in Table 3, 5 epochs of warmup does not bring in further improvement.

**CenterNorm.** We compare CenterNorm against no-Norm (as in ReZero [3]) and the standard LayerNorm. Results show that: 1) Lipsformer-CSwin with LayerNorm becomes unstable and does not converge, but with CenterNorm and no-Norm, LipsFormer-CSwin can successfully converge; 2) Using CenterNorm significantly outperforms no-Norm by 1.3%.

| Warmup | Tiny | | Normalization | Tiny | | Initialization | Tiny | | Attention | Tiny |
|---|---|---|---|---|---|---|---|---|---|---|
| No | 81.6 | | LayerNorm | Not converge | | Truncated Normal | 81.3 | | Dot Product | Diverged |
| Yes | 81.2 | | No Norm | 80.3 | | Xavier | 81.6 | | L2 Distance Attn | 81.3 |
| | | | CenterNorm | 81.6 | | Spectral | 81.6 | | SCSA | 81.6 |

| Alpha | Tiny | | Conv Blocks | Tiny | | Drop Ratio | Tiny |
|---|---|---|---|---|---|---|---|
| $\alpha = 0.1$ | 81.4 | | no Conv | 80.7 | | $p = 0.0$ | 81.3 |
| $\alpha = 0.2$ | 81.6 | | Conf. B | 81.2 | | $p = 0.1$ | 81.6 |
| $\alpha = 0.3$ | 81.3 | | Conf. C | 81.6 | | $p = 0.2$ | 81.6 |
| $\alpha = 0.4$ | Diverged | | Conf. D | 81.6 | | | |

TABLE 3: Ablation study on key components of LipsFormer. "Not converge" means training loss oscillates without converging, and "Diverged" means the loss explodes because of "NaN" or "Inf".

**Spectral Initialization.** We compare LipsFormer-CSwin results with spectral initialization against truncated normal and Xavier initialization. We find that LipsFormer-CSwin with any of the three initializations converges. Spectral initialization and Xavier initialization slightly outperforms truncated normal initialization, but spectral initialization has a better Lipschitz interpretability than Xavier initialization.

**Scaled Cosine Similarity Attention.** To validate the effectiveness of scaled cosine similarity attention, we compare it with the standard dot-product attention and the L2 distance attention [28]. We find that the standard dot-product self-attention leads to forward value explosion, but the scaled cosine similarity attention works well under Lipschitz guarantee. Meanwhile, SCSA works better than the L2 distance attention.

**Impact of the Residual Weight $\alpha$.** As detailed in 4.1.3, the weight of residual path $\alpha$ has a substantial influence on the upper bound of LipsFormer's Lipschitz constant. We evaluate different choices of $\alpha$, and find that with a large $\alpha$ initialization value, network either does not converge or diverges quickly. This validates that deeper networks need a smaller $\alpha$.

**Convolution Blocks.** In LipsFormer-CSwin, we use two depth-wise convolutions (dwc) and one point-wise convolution (pwc). We evaluate four different convolution configurations: A) no convolution; B) one dwc; C) dwc + pwc; and D) dwc + pwc + dwc. Table 3 shows that one dwc increases LipsFormer's accuracy by 0.5%, one dwc + one pwc further improves its performance by 0.4%, adding more convolutions saturates performance gains.

**DropPath Ratio.** In Appendix J, we show that DropPath effectively decreases the upper bound of a network's Lipschitz constant, making training process more stable. The results in Table 3 show that reasonable DropPath can effectively improve training performance.

To summarize, CenterNorm, scaled cosine similarity attention, and convolution blocks all contribute positively to LipsFormer-CSwin's superior performance. Weighted residual shortcut with small $\alpha$, reasonable DropPath ratio $p$ and spectral initialization are effective in stabilizing LipsFormer-CSwin by constraining its Lipschitz constant.

## 6 CONCLUSION

In this paper, we present a Lipschitz continuous Transformer, called LipsFormer, to pursue a more stable training process by enforcing the Lipschitz continuity of the whole network. We analyze key components of Transformer and replace the ones violating Lipschitz continuity by introducing CenterNorm, scaled cosine similarity attention, and spectral initialization. LipsFormer also uses weighted residual shortcut and DropPath to further decrease the upper bound of its Lipschitz constant. Finally, we derive an upper bound of the Lipschitz constant of a LipsFormer network architecture. We empirically validate the effectiveness of LipsFormer-Swin and LipsFormer-CSwin, based on Swin Transformer and CSwin individually, on ImageNet 1K classification with state-of-the-art performance for model variants of different parameter sizes. The analysis of the Lipschitz continuity of a network is general. We look forward to extending it to a broader class of models and application areas, including multi-modal model and natural language processing. We also hope future works will discuss the Lipschitz continuity of LipsFormer in the backward process in depth.

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

## A  APPENDIX

### A.1  LIPSCHITZ CONSTANT OF COMMON ACTIVATION FUNCTIONS

In Figure 2 we plot common non-linear activation functions in neural networks: Sigmoid, Tanh, ReLU and GELU. According to [25], GeLU can be approximated by $\mathrm{GeLU}(x) \approx x\,\mathrm{sigmoid}(1.702x)$. According to Lemma 1, the Lipschitz constants of Sigmoid, Tanh, ReLU and GELU are $\frac{1}{4}$, 1, 1, 1.0998 respectively.

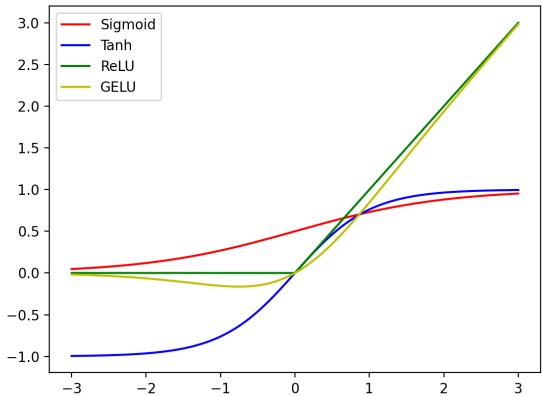

FIGURE 2: Sigmoid, Tanh, ReLU and GELU activation function.

### A.2  MULTI-HEAD ATTENTION

For a $K$-head attention, we have the $i$-th attention, $i \in \{1, ..., K\}$ defined as,
$$\boldsymbol{h}_i(\boldsymbol{x}, \boldsymbol{W}_i) = \mathrm{Attn}_i(\boldsymbol{X}, \boldsymbol{W}_i^Q, \boldsymbol{W}_i^K, \boldsymbol{W}_i^V),$$
where $\boldsymbol{W}_i$ is the set of projection weight matrices $(\boldsymbol{W}_i^Q, \boldsymbol{W}_i^K, \boldsymbol{W}_i^V)$.

Multi-head attention simply concatenates different attention results,
$$\boldsymbol{h}(\boldsymbol{x}, \boldsymbol{W}) = [\boldsymbol{h}_1(\boldsymbol{x}, \boldsymbol{W}_1);\ \boldsymbol{h}_2(\boldsymbol{x}, \boldsymbol{W}_2);\ ...;\ \boldsymbol{h}_K(\boldsymbol{x}, \boldsymbol{W}_K)].$$

According to the Lipschitz definition, we have,
$$\|\boldsymbol{h}(\boldsymbol{x}_1, \boldsymbol{W}) - \boldsymbol{h}(\boldsymbol{x}_2, \boldsymbol{W})\| \leq (\mathrm{Lip}(\boldsymbol{h}_1(\boldsymbol{W}_1)) + \mathrm{Lip}(\boldsymbol{h}_2(\boldsymbol{W}_2)) + ... + \mathrm{Lip}(\boldsymbol{h}_K(\boldsymbol{W}_K)))\|\boldsymbol{x}_1 - \boldsymbol{x}_2\|.$$

## B  NETWORK ARCHITECTURE AND CONFIGURATIONS

The overall architecture of LipsFormer-CSwin is shown in Figure 3. For patch embedding and patch merging, we use non-overlapped convolution as in Swin Transformer. Following CSwin Transformer, we use the same cross-shaped window when computing attention results and also the same Locally enhanced Positional Encoding (LePE).

The configurations of Lipsformer-CSwin are based on CSwin Transformer and Table 4 summarizes three variants of Lipsformer-CSwin. LipsFormer-CSwin-T and LipsFormer-CSwin-S only varies in the number of LipsFormer-CSwin blocks. LipsFormer-CSwin-S/B share the same depth configuration but varies in hidden layer channel size.

Similar to LipsFormer-CSwin, we also build LipsFormer based on Swin Transformer [35]. Here, we term it as LipsFormer-Swin. We create five versions of LipsFormer-Swin, and the detailed configurations are shown in Table 5.

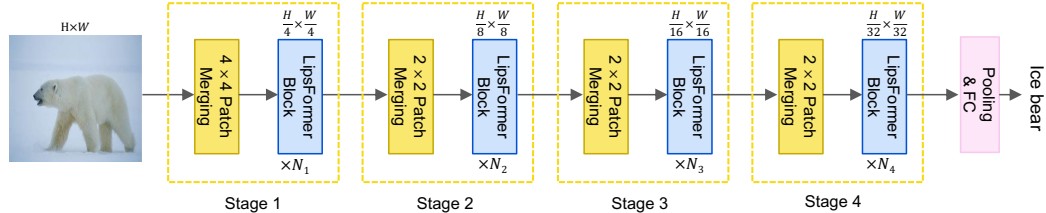

FIGURE 3: Illustration of the overall LipsFormer architecture.

TABLE 4: Details of LipsFormer-CSwin model variants.

| Model | Channel | Number of Blocks | Num. of Params. |
|---|---|---|---|
| LipsFormer-CSwin-T | 64 | [1, 2, 21, 1] | 24M |
| LipsFormer-CSwin-S | 64 | [2, 4, 32, 2] | 38M |
| LipsFormer-CSwin-B | 96 | [2, 4, 32, 2] | 83M |

## C  TRAINING DETAILS

In Table 6 we provide the ImageNet 1K training details used for producing the main results in Table 2. All LipsFormer variants use the same training hyperparameters, except for DropPath ratio, weight decay, learning rate and EMA. All the models are implemented with PyTorch, and trained on NVIDIA Tesla A100 GPUs. We do *not* use learning rate warmup in all experiments.

## D  EXPERIMENTS OF LIPSFORMER-SWIN

We evaluate the Tiny, Small, Base and Large versions of LipsFormer-Swin on the ImageNet-1K, and compare our results with their corresponding counterpart Swin Transformer. The results are shown in Table 7.

We have the following two findings from Table 7,

- the proposed LipsFormer-Swin consistently outperforms its counterpart Swin Transformer. Specifically, LipsFormer-Swin-T improves Swin-T by 1.5%.

- LipsFormer-Swin-L shows obvious overfitting on ImageNet-1K, and performs worst than LipsFormer-Swin-B. According to our observation in the training process, the training loss (around 2.2) of LipsFormer-Swin-L is much smaller than that (around 2.5) of LipsFormer-Swin-B, but the test accuracy is lower.) We also observe that in some github discussion issues, some people[1] also find that the original Swin-L cannot outperform Swin-B if only training on ImageNet-1K.

Since LipsFormer-Swin-L has shown overfitting on ImageNet-1K, we do not report the performance of LipsFormer-Swin-L++ on the table. In the future, we will train it on a larger scale of data to test its fitting ability. On a single A100-40GB GPU, with a batch size fixed to 256 and a mixed precision,

---

[1]https://github.com/microsoft/Swin-Transformer/issues/261

TABLE 5: Details of LipsFormer-Swin model variants.

| Model | Channel | Num. of Blocks | Num. of Params. |
|---|---|---|---|
| LipsFormer-Swin-T | 96 | [2, 2, 6, 2] | 31 |
| LipsFormer-Swin-S | 96 | [2, 2, 18, 2] | 54 |
| LipsFormer-Swin-B | 128 | [2, 2, 18, 2] | 96 |
| LipsFormer-Swin-L | 192 | [2, 2, 18, 2] | 214 |
| LipsFormer-Swin-L++ | 288 | [2, 2, 18, 2] | 526 |

TABLE 6: Hyperparameters for the models. $L_t, L_s, L_b, L_\ell$ are the number of residual blocks in tiny, small, base and large.

| Hyperparameters | Tiny | Small | Base | Large |
|---|---|---|---|---|
| Warmup steps | **0** | **0** | **0** | **0** |
| Optimizer | AdamW | AdamW | AdamW | AdamW |
| DropPath ratio | 0.2 | 0.4 | 0.5 | 0.5 |
| $\alpha$ in residual as in Eq. 4 | $\frac{1}{L_t}$ | $\frac{1}{L_s}$ | $\frac{1}{L_b}$ | $\frac{1}{L_\ell}$ |
| $\tau$ in cosine attention as in Eq. 4.1.2 | 12 | 12 | 12 | 12 |
| Learning rate | 2e-3 | 2e-3 | 1e-3 | 1e-3 |
| Learning rate scheduler | cosine | cosine | cosine | cosine |
| Adam $\epsilon$ | 1e-8 | 1e-8 | 1e-8 | 1e-8 |
| Adam $\beta$ | $(0.9, 0.999)$ | $(0.9, 0.999)$ | $(0.9, 0.999)$ | $(0.9, 0.99)$ |
| Label smoothing | 0.1 | 0.1 | 0.1 | 0.1 |
| RandAugment | (9, 0.5) | (9, 0.5) | (9, 0.5) | (9, 0.5) |
| Mixup | 0.8 | 0.8 | 0.8 | 0.8 |
| Cutmix | 1.0 | 1.0 | 1.0 | 1.0 |
| Training epochs | 300 | 300 | 300 | 300 |
| Gradient clipping | 0.0 | 0.0 | 0.0 | 0.0 |
| Weight decay | 0.05 | 0.05 | 0.1 | 0.1 |
| EMA | 0.99984 | 0.99984 | 0.99992 | 0.99992 |

TABLE 7: Comparison of our LipsFormer-Swin with its corresponding counterpart Swin Transformer with input resolution $224^2$ on ImageNet-1K classification.

| Method | Param. (M) | FLOPs (G) | Acc. | Method | Param. (M) | FLOPs (G) | Acc. |
|---|---|---|---|---|---|---|---|
| Swin-T | 29 | 4.5 | 81.3 | LipsFormer-Swin-T | 31 | 5.0 | 82.7 |
| Swin-S | 50 | 8.7 | 83.0 | LipsFormer-Swin-S | 54 | 9.7 | 83.5 |
| Swin-B | 88 | 15.4 | 83.5 | LipsFormer-Swin-B | 96 | 17.0 | 84.0 |
| Swin-L | 190.7 | 34.5 | N/A | LipsFormer-Swin-L | 214 | 37.7 | 83.5 |

the throughput and peak memory are 963 im/s and 5483 MB for LipsFormer-Swin-T, and 433 im/s and 7415 MB for LipsFormer-Swin-B.

# E  OVERFITTING EVALUATION

Following DeiT III [47], we also evaluate our method on ImageNet-v2 [43] and ImageNet-real [4] data sets. As pointed out by [48], to test how the method performs in a nearby setting without any finetuning is a good way to assess overfitting. We directly apply the obtained models trained on the original ImageNet data set onto these two data sets. The results are shown in Table 8.

We can see that from Table 8, our method that works well on the original ImageNet data set consistently performs well on the ImageNet-v2 and ImageNet-real data sets. This observation fully validates the generalization ability of the proposed method.

# F  TRAINING CURVES

In Figure 4, we show the training curves of the training losses and the classification accuracies of LipsFormer-Swin-T and LipsFormer-Swin-B. We can find LipsFormer-Swin-B can fit the training data better than LipsFormer-Swin-T because the loss of LipsFormer-Swin-B is much lower than that of LipsFormer-Swin-T.

TABLE 8: Details of LipsFormer-CSwin model variants. All results except our LipsFormer-CSwin are taken from DeiT III [47]

| Model | Params (M) | Flops (G) | val | real | v2 |
|---|---|---|---|---|---|
| ViT-S | 22.0 | 4.6 | 80.4 | 86.1 | 69.7 |
| PiT-S | 23.5 | 2.9 | 80.4 | 86.1 | 69.2 |
| TNT-S | 23.8 | 5.2 | 81.4 | 87.2 | 70.6 |
| ConViT-S | 27.8 | 5.8 | 81.3 | 87.0 | 70.3 |
| Swin-S | 49.6 | 8.7 | 82.1 | 86.9 | 70.7 |
| **LipsFormer-CSwin-T** | 24 | 4.7 | **83.5** | **88.0** | **73.2** |
| ViT-B | 86.6 | 17.6 | 83.1 | 87.7 | 72.6 |
| PiT-B | 73.8 | 12.5 | 82.4 | 86.8 | 72.0 |
| TNT-B | 65.6 | 14.1 | 82.9 | 87.6 | 72.2 |
| ConViT-B | 86.5 | 17.5 | 82.0 | 86.7 | 71.3 |
| Swin-B | 87.8 | 15.4 | 82.2 | 86.7 | 70.7 |
| CaiT-B12 | 100.0 | 18.2 | 83.3 | 87.7 | 73.3 |
| **LipsFormer-CSwin-B** | 83 | 16.3 | **84.6** | **88.6** | **74.5** |

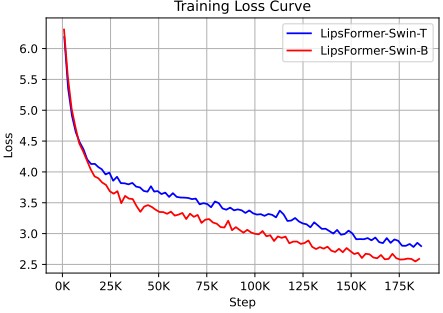
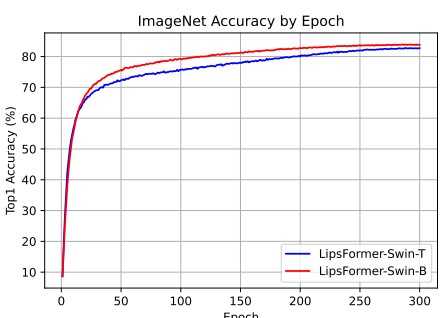

FIGURE 4: Training curves of LipsFormer-Swin-T and LipsFormer-Swin-B. Left: training loss along with epochs. Right: classification accuracy along with epochs.

## G PARAMETER VARIATIONS ALONG WITH TRAINING EPOCHS

In Figure 5, we show the variations of the $\alpha$ along with the training epochs. Our statistic is based on LipsFormer-Swin-T model. We statistic the mean and standard variance of the absolute value of the $\alpha$. We select one set of $\alpha$ from each stage. We find that from Figure 5, the mean value of the absolute value of the $\alpha$ first grows and then tend to stabilize at a value.

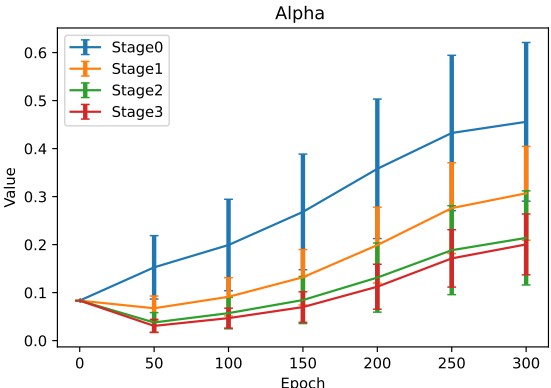

FIGURE 5: Variation curve of $\boldsymbol{\alpha}$ along with training epochs. Each line denotes one set of $\boldsymbol{\alpha}$ in one stage. We show the mean and standard variance of the absolute value of the $\boldsymbol{\alpha}$.

## H  PROOF OF THEOREM 1

In this subsection, we derive the Lipschitz constant upper bound for the scaled cosine similarity attention (SCSA).

First, we list some useful notations and identities for deriving the Jacobians of attention computation.

$$
\boldsymbol{X} = \begin{bmatrix} - & \mathbf{x}_1^\top & - \\ & \vdots & \\ - & \mathbf{x}_N^\top & - \end{bmatrix} \in \mathbb{R}^{N \times D}.
$$

For column vectors $\mathbf{u}, \mathbf{z} \in \mathbb{R}^N$ the chain rule has:

$$
\frac{\partial}{\partial \mathbf{x}} \left[ \mathbf{u}^\top \mathbf{z} \right] = \mathbf{u}^\top \frac{\partial \mathbf{z}}{\partial \mathbf{x}} + \mathbf{z}^\top \frac{\partial \mathbf{u}}{\partial \mathbf{x}}.
$$

The standard dot-product attention is defined as,

$$
\mathrm{DP}(\boldsymbol{X}, \boldsymbol{W}^Q, \boldsymbol{W}^K, \boldsymbol{W}^V) := \mathrm{softmax}\left( \frac{\boldsymbol{X}\boldsymbol{W}^Q \left(\boldsymbol{X}\boldsymbol{W}^K\right)^\top}{\sqrt{D/H}} \right) \boldsymbol{X}\boldsymbol{W}^V
$$

$$
= \boldsymbol{P}\boldsymbol{X}\boldsymbol{W}^V.
$$

In [28], Kim *et al.*proved that the standard dot-product attention is not Lipschitz continuous, and proposed L2-distance attention which is Lipschitz continuous conditioning on $\boldsymbol{W}^Q = \boldsymbol{W}^K$. But enforcing the equality of $\boldsymbol{W}^Q$ and $\boldsymbol{W}^K$ limits the expressiveness of the Transformer and degrades training performance empirically.

Our scaled cosine similarity attention is defined as,

$$
\mathrm{SCSA}(\boldsymbol{X}, \boldsymbol{W}^Q, \boldsymbol{W}^K, \boldsymbol{W}^V, \nu, \tau) = \nu \boldsymbol{P}\boldsymbol{V},
$$
$$
\text{where } \boldsymbol{P} = \mathrm{softmax}\left( \tau \boldsymbol{Q}\boldsymbol{K}^\top \right),
$$

(6)

$\nu$ and $\tau$ are predefined or learnable scalars.

The definitions of $\boldsymbol{Q}, \boldsymbol{K}, \boldsymbol{V}$ are as follows,

$$
\boldsymbol{Q} = \begin{bmatrix} - & \mathbf{q}_1^\top & - \\ & \vdots & \\ - & \mathbf{q}_N^\top & - \end{bmatrix} \in \mathbb{R}^{N \times D}, \quad \boldsymbol{K} = \begin{bmatrix} - & \mathbf{k}_1^\top & - \\ & \vdots & \\ - & \mathbf{k}_N^\top & - \end{bmatrix} \in \mathbb{R}^{N \times D}, \quad \boldsymbol{V} = \begin{bmatrix} - & \mathbf{v}_1^\top & - \\ & \vdots & \\ - & \mathbf{v}_N^\top & - \end{bmatrix} \in \mathbb{R}^{N \times D}.
$$

For each input $\mathbf{x}_i$, the projected $\mathbf{q}_i$, $\mathbf{k}_i$, $\mathbf{v}_i$ are defined as,

$$\mathbf{q}_i = \frac{(\mathbf{x}_i^\top \boldsymbol{W}^Q)^\top}{\sqrt{\|\mathbf{x}_i^\top \boldsymbol{W}^Q\|^2 + \epsilon}}, \quad \mathbf{k}_j = \frac{(\mathbf{x}_j^\top \boldsymbol{W}^K)^\top}{\sqrt{\|\mathbf{x}_j^\top \boldsymbol{W}^K\|^2 + \epsilon}}, \quad \mathbf{v}_j = \frac{(\mathbf{x}_j^\top \boldsymbol{W}^V)^\top}{\sqrt{\|\mathbf{x}_j^\top \boldsymbol{W}^V\|^2 + \epsilon}}.$$

where $\epsilon$ is a small smoothing factor to guarantee that the definition of cosine similarity is valid everywhere.

By taking partial derivatives, we have the following Jacobian matrices,

$$\widetilde{\boldsymbol{Q}_i} = \frac{\partial \mathbf{q}_i}{\partial \mathbf{x}_i} = \frac{1}{\sqrt{\|\mathbf{x}_i^\top \boldsymbol{W}^Q\|^2 + \epsilon}} (\boldsymbol{I} - \frac{\boldsymbol{W}^{Q\top} \mathbf{x}_i \mathbf{x}_i^\top \boldsymbol{W}^Q}{\|\mathbf{x}_i^\top \boldsymbol{W}^Q\|^2 + \epsilon}) \boldsymbol{W}^{Q\top}, \tag{7}$$

$$\widetilde{\boldsymbol{K}_j} = \frac{\partial \mathbf{k}_j}{\partial \mathbf{x}_j} = \frac{1}{\sqrt{\|\mathbf{x}_j^\top \boldsymbol{W}^K\|^2 + \epsilon}} (\boldsymbol{I} - \frac{\boldsymbol{W}^{K\top} \mathbf{x}_j \mathbf{x}_j^\top \boldsymbol{W}^K}{\|\mathbf{x}_j^\top \boldsymbol{W}^K\|^2 + \epsilon}) \boldsymbol{W}^{K\top}, \tag{8}$$

$$\widetilde{\boldsymbol{V}_j} = \frac{\partial \mathbf{v}_j}{\partial \mathbf{x}_j} = \frac{1}{\sqrt{\|\mathbf{x}_j^\top \boldsymbol{W}^V\|^2 + \epsilon}} (\boldsymbol{I} - \frac{\boldsymbol{W}^{V\top} \mathbf{x}_j \mathbf{x}_j^\top \boldsymbol{W}^V}{\|\mathbf{x}_j^\top \boldsymbol{W}^V\|^2 + \epsilon}) \boldsymbol{W}^{V\top}. \tag{9}$$

The attention matrix $\boldsymbol{P}$ is defined as,

$$\boldsymbol{P} := \text{softmax} \begin{pmatrix} \tau \mathbf{q}_1^\top \mathbf{k}_1 & \tau \mathbf{q}_1^\top \mathbf{k}_2 & \dots & \tau \mathbf{q}_1^\top \mathbf{k}_n \\ \tau \mathbf{q}_2^\top \mathbf{k}_1 & \tau \mathbf{q}_2^\top \mathbf{k}_2 & \dots & \tau \mathbf{q}_2^\top \mathbf{k}_n \\ \vdots & \vdots & \ddots & \vdots \\ \tau \mathbf{q}_n^\top \mathbf{k}_1 & \tau \mathbf{q}_n^\top \mathbf{k}_2 & \dots & \tau \mathbf{q}_n^\top \mathbf{k}_n \end{pmatrix}.$$

We can rewrite our SCSA attention in Eq. 6 as,

$$f(\boldsymbol{X}) = \nu \boldsymbol{P} \boldsymbol{V} = \nu \, \text{softmax} \left( \tau \boldsymbol{Q} \boldsymbol{K}^\top \right) \boldsymbol{V} = \begin{bmatrix} f_1(\boldsymbol{X})^\top \\ \vdots \\ f_N(\boldsymbol{X})^\top \end{bmatrix} \in \mathbb{R}^{N \times D}. \tag{10}$$

For simplification we focus on derivations for single-head attention, mutli-head attention requires only minor modifications for concatenating attention results for each head as discussed in A.2 . The Jacobian matrix for SCSA can be written as,

$$\boldsymbol{J}_f = \begin{bmatrix} \boldsymbol{J}_{11} & \cdots & \boldsymbol{J}_{1N} \\ \vdots & \ddots & \vdots \\ \boldsymbol{J}_{N1} & \cdots & \boldsymbol{J}_{NN} \end{bmatrix} \in \mathbb{R}^{ND \times ND},$$

$$\text{where } \boldsymbol{J}_{ij} = \frac{\partial f_i(\boldsymbol{X})}{\partial \mathbf{x}_j} \in \mathbb{R}^{D \times D}.$$

By taking partial derivatives we can show that,

$$\boldsymbol{J}_{ij} = \nu \tau \boldsymbol{V}^\top \boldsymbol{P}^{(i)} \left[ \boldsymbol{E}_{ji} \boldsymbol{Q} \widetilde{\boldsymbol{K}_j}^\top + \delta_{ij} \boldsymbol{K} \widetilde{\boldsymbol{Q}_j}^\top \right] + \nu P_{ij} \widetilde{\boldsymbol{V}_j}, \tag{11}$$

where $\boldsymbol{E}_{ij} \in R^{N \times N}$ is a binary matrix with zeros everywhere except the (i, j)-th entry, $\delta_{ij}$ is the Kronecker delta, and the Jacobian of the softmax is well-known as below,

$$\boldsymbol{P}^{(i)} := \operatorname{diag}\left(\boldsymbol{P}_{i:}\right) - \boldsymbol{P}_{i:}^{\top} \boldsymbol{P}_{i:} = \begin{bmatrix} P_{i1}\left(1 - P_{i1}\right) & -P_{i1}P_{i2} & \cdots & -P_{i1}P_{iN} \\ -P_{i2}P_{i1} & P_{i2}\left(1 - P_{i2}\right) & \cdots & -P_{i2}P_{iN} \\ \vdots & \vdots & \ddots & \vdots \\ -P_{iN}P_{i1} & -P_{iN}P_{i2} & \cdots & P_{iN}\left(1 - P_{iN}\right) \end{bmatrix}. \quad (12)$$

When $i = j$, we have,

$$\boldsymbol{J}_{ii} = \nu\tau\boldsymbol{V}^{\top}\boldsymbol{P}^{(i)}\left[\boldsymbol{E}_{ii}\boldsymbol{Q}\widetilde{\boldsymbol{K}_i}^{\top} + \boldsymbol{K}\widetilde{\boldsymbol{Q}_i}^{\top}\right] + \nu P_{ii}\widetilde{\boldsymbol{V}_i}. \quad (13)$$

When $i \neq j$, we have,

$$\boldsymbol{J}_{ij} = \nu\tau\boldsymbol{V}^{\top}\boldsymbol{P}^{(i)}\boldsymbol{E}_{ji}\boldsymbol{Q}\widetilde{\boldsymbol{K}_j}^{\top} + \nu P_{ij}\widetilde{\boldsymbol{V}_j}. \quad (14)$$

**Lemma 3.** *The scaled cosine similarity attention (SCSA) is Lipschitz continuous if and only if $\boldsymbol{W}^Q, \boldsymbol{W}^K, \boldsymbol{W}^V$ have bounded norm.*

*Sketch Proof.* Our key observation is that most of the terms in $J_{ii}$ and $J_{ij}$ have bounded norm: $\nu$ and $\tau$ are scalars; $\boldsymbol{Q}, \boldsymbol{K}, \boldsymbol{V}$ are normalized so all elements are less than or equal to 1; $\boldsymbol{E}_{ij}$ has zeros everywhere except the (i,j)-th entry; $\boldsymbol{P}$ is an attention matrix with all elements within $[0, 1]$ so all elements in $\boldsymbol{P}^{(i)}$ are bounded by $[-0.25, 0.25]$. Taking a closer look at $\widetilde{\boldsymbol{Q}_i}, \widetilde{\boldsymbol{K}_i}, \widetilde{\boldsymbol{V}_i}$ as shown in Eq. 7, Eq. 8 and Eq. 9, they are bounded as long as $\boldsymbol{W}^Q, \boldsymbol{W}^K, \boldsymbol{W}^V$ are bounded. Consequently the final product of $J_{ii}$ and $J_{ij}$ have bounded norm if $\boldsymbol{W}^Q, \boldsymbol{W}^K, \boldsymbol{W}^V$ have bounded norm.

### H.1 UPPER BOUND ON LIP$\infty$ FOR SCSA

Let us review some basic definitions for matrix norm. Suppose we have matrices $\boldsymbol{A} \in \mathbb{R}^{N \times D}$, and $\boldsymbol{B} \in \mathbb{R}^{N \times D}$. Then, we have:

$$\|\boldsymbol{A}\|_\infty = \max_{1 \leq i \leq N} \sum_{j=1}^{D} |\boldsymbol{A}_{ij}|,$$

$$\|\boldsymbol{A}\|_2 = \sqrt{\lambda_{\max}\left(\boldsymbol{A}^*\boldsymbol{A}\right)} = \sigma_{\max}(\boldsymbol{A}).$$

We also have the following inequalities,

$$\|\boldsymbol{A}\boldsymbol{B}^{\top}\| \leq \|\boldsymbol{A}\|\|\boldsymbol{B}^{\top}\|, \|\boldsymbol{A} + \boldsymbol{B}\| \leq \|\boldsymbol{A}\| + \|\boldsymbol{B}\| \text{ and } \|[\boldsymbol{A}_1, \ldots, \boldsymbol{A}_N]\| \leq \sum_i \|\boldsymbol{A}_i\|.$$

$$\|\boldsymbol{A}\|_2 = \sigma_{\max}(\boldsymbol{A}) \leq \|\boldsymbol{A}\|_{\mathrm{F}} = \left(\sum_{i=1}^{N}\sum_{j=1}^{D} |\boldsymbol{A}_{ij}|^2\right)^{\frac{1}{2}} = \left(\sum_{k=1}^{\min(N,D)} \sigma_k^2\right)^{\frac{1}{2}}, \quad (15)$$

where $\|\cdot\|_{\mathrm{F}}$ is the Frobenius norm. Equality holds if and only if matrix $\boldsymbol{A}$ is a rank-one matrix or a zero matrix.

According to the above inequalities, we have

$\|[\boldsymbol{J}_{i1}, \ldots, \boldsymbol{J}_{iN}]\|_\infty$

$\leq \|\boldsymbol{J}_{ii}\|_\infty + \sum_{j \neq i}\|\boldsymbol{J}_{ij}\|_\infty$

$\leq \nu\tau\|\boldsymbol{V}^{\top}\|_\infty\|\boldsymbol{P}^{(i)}\|_\infty\left[\|\boldsymbol{E}_{ii}\|_\infty\|\boldsymbol{Q}\|_\infty\|\widetilde{\boldsymbol{K}_i}^{\top}\|_\infty + \|\boldsymbol{K}\|_\infty\|\widetilde{\boldsymbol{Q}_i}^{\top}\|_\infty\right] + \nu\|P_{ii}\|_\infty\|\widetilde{\boldsymbol{V}_i}\|_\infty +$

$\sum_{j \neq i}\nu\tau\|\boldsymbol{V}^{\top}\|_\infty\|\boldsymbol{P}^{(i)}\|_\infty\|\boldsymbol{E}_{ji}\|_\infty\|\boldsymbol{Q}\|_\infty\|\widetilde{\boldsymbol{K}_j}^{\top}\|_\infty + \nu\|P_{ij}\|_\infty\|\widetilde{\boldsymbol{V}_j}\|_\infty$

$$(16)$$

We can compute the $L_2$ norm Lipschitz constant by replacing the $L_\infty$ norm in the above equation with $L_2$ norm.

With simple derivations we list $\|\cdot\|_\infty$ for each term in 16:

$\|\boldsymbol{V}^\top\|_\infty = \max_{1\le i\le D} \sum_{j=1}^N \|\boldsymbol{V}^\top{}_{ij}\| \le N$

$\|\boldsymbol{P}^{(i)}\|_\infty = \max_{1\le i\le N} \sum_{j=1}^N \|\boldsymbol{P}^{(i)}{}_{ij}\| = \max_{1\le i\le N} 2(P_{ii} - P_{ii}^2) \le \frac{1}{2}$

$\|\boldsymbol{E}_{ii}\|_\infty = 1$

$\|\boldsymbol{Q}\|_\infty = \max_{1\le i\le N} \sum_{j=1}^D \|\boldsymbol{Q}_{ij}\| \le \sqrt{D}$

$$\|\widetilde{\boldsymbol{K}_j}^\top\|_\infty \le \epsilon^{-\frac{1}{2}} \times \|\boldsymbol{W}^K\|_\infty \times 2 \tag{17}$$

*Proof. for Equation* 17

$$
\begin{aligned}
\|\widetilde{\boldsymbol{K}_j}^\top\|_\infty &= \left\| \left[ \frac{1}{\sqrt{\|\mathbf{x}_j{}^\top \boldsymbol{W}^K\|^2 + \epsilon}} \left(\boldsymbol{I} - \frac{\boldsymbol{W}^{K\top}\mathbf{x}_j\mathbf{x}_j{}^\top \boldsymbol{W}^K}{\|\mathbf{x}_j{}^\top \boldsymbol{W}^K\|^2 + \epsilon}\right) \boldsymbol{W}^{K\top} \right]^\top \right\|_\infty \\
&\le \epsilon^{-\frac{1}{2}} \times \|\boldsymbol{W}^K\|_\infty \times \left\|\left(\boldsymbol{I} - \frac{\boldsymbol{W}^{K\top}\mathbf{x}_j\mathbf{x}_j{}^\top \boldsymbol{W}^K}{\|\mathbf{x}_j{}^\top \boldsymbol{W}^K\|^2 + \epsilon}\right)\right\|_\infty \\
&\le 2 \times \epsilon^{-\frac{1}{2}} \times \|\boldsymbol{W}^K\|_\infty \qquad\qquad\qquad \square
\end{aligned}
$$

$\|\widetilde{\boldsymbol{K}_i}^\top\|_\infty = \|\widetilde{\boldsymbol{K}_j}^\top\|_\infty < \epsilon^{-\frac{1}{2}} \times \|\boldsymbol{W}^K\|_\infty \times 2$

$\|\boldsymbol{K}\|_\infty = \sqrt{D}$

$\|\widetilde{\boldsymbol{Q}_i}^\top\|_\infty \le 2 \times \epsilon^{-\frac{1}{2}} \times \|\boldsymbol{W}^Q\|_\infty$, similar to Equation 17

$\|P_{ii}\|_\infty = \|P_{ij}\|_\infty = 1$

$\|\boldsymbol{E}_{ji}\|_\infty = 1$

$\|\widetilde{\boldsymbol{V}_j}\|_\infty \le 2 \times \epsilon^{-\frac{1}{2}} \times \|\boldsymbol{W}^{V\top}\|_\infty$, similar to Equation 17,

According to 16, the Lip$\infty$ constant of the scaled cosine similarity attention (SCSA) is:

$$
\begin{aligned}
\text{Lip}(SCSA)_\infty \le{}& \nu \times \tau \times N \times \frac{1}{2} \times \left[ 1 \times \sqrt{D} \times \epsilon^{-\frac{1}{2}} \times 2 \times \|\boldsymbol{W}^K\|_\infty + \sqrt{D} \times \epsilon^{-\frac{1}{2}} \times 2 \times \|\boldsymbol{W}^Q\|_\infty \right] + \\
& \nu \times 1 \times \epsilon^{-\frac{1}{2}} \times 2 \times \|\boldsymbol{W}^{V\top}\|_\infty + \\
& (N-1)\left[ \nu \times \tau \times N \times \frac{1}{2} \times 1 \times \sqrt{D} \times \epsilon^{-\frac{1}{2}} \times 2 \times \|\boldsymbol{W}^K\|_\infty + \nu \times 1 \times \epsilon^{-\frac{1}{2}} \times 2 \times \|\boldsymbol{W}^{V\top}\|_\infty \right].
\end{aligned}
$$

After merging and rearranging the terms,

$$
\begin{aligned}
\text{Lip}(SCSA)_\infty ={}& \nu\tau N\sqrt{D}\epsilon^{-\frac{1}{2}} \left[\|\boldsymbol{W}^K\|_\infty + \|\boldsymbol{W}^Q\|_\infty\right] + 2\nu\epsilon^{-\frac{1}{2}}\|\boldsymbol{W}^{V\top}\|_\infty + \\
& (N-1)\left[\nu\tau N\sqrt{D}\epsilon^{-\frac{1}{2}}\|\boldsymbol{W}^K\|_\infty + 2\nu\epsilon^{-\frac{1}{2}}\|\boldsymbol{W}^{V\top}\|_\infty\right] \\
={}& N^2\sqrt{D}\nu\tau\epsilon^{-\frac{1}{2}}\|\boldsymbol{W}^K\|_\infty + N\sqrt{D}\nu\tau\epsilon^{-\frac{1}{2}}\|\boldsymbol{W}^Q\|_\infty + 2N\nu\epsilon^{-\frac{1}{2}}\|\boldsymbol{W}^{V\top}\|_\infty
\end{aligned}
$$

## H.2 Upper Bound on Lip$_2$ For SCSA

Correspondingly, we list $\|\cdot\|_2$ for each term in 16:

$$\|\boldsymbol{V}^\top\|_2 \leq \left(\sum_{i=1}^N \sum_{j=1}^D |V_{ij}|^2\right)^{\frac{1}{2}} = \left(\sum_{j=1}^N 1\right)^{\frac{1}{2}} = \sqrt{N}$$

$$\|\boldsymbol{P}^{(i)}\|_2 \leq \frac{N-1}{N} \tag{18}$$

*Proof of Equation 18*

According to Eq 12, $\boldsymbol{P}^{(i)}$ is a semi-definite matrix, thus its ordered eigenvalues $\lambda_1 \geq \lambda_2 \geq, \ldots, \geq \lambda_N \geq 0$, and $\sum_{i=1}^N \lambda_i = \text{tr}(\boldsymbol{P}^{(i)}) = \sum_j^N \boldsymbol{P}_{jj}^{(i)} \leq (\sum_{j=1}^N \frac{1}{N}\frac{N-1}{N}) = \frac{N-1}{N}$.

According to 15, $\|\boldsymbol{P}^{(i)}\|_2 = (\sum_{i=1}^N \lambda_i^2)^{\frac{1}{2}} \leq (\sum_{i=1}^N \lambda_i)^{2 \times \frac{1}{2}} \leq \frac{N-1}{N}$ $\qquad\qquad\square$

$\|\boldsymbol{E}_{ii}\|_2 = 1$

$\|\boldsymbol{Q}\|_2 \leq \sqrt{N}$

$\|\widetilde{\boldsymbol{K}_j}^\top\|_2 \leq 2 \times \epsilon^{-\frac{1}{2}} \times \|\boldsymbol{W}^K\|_2,$

$\|\boldsymbol{K}\|_2 \leq \sqrt{N}$

$\|\widetilde{\boldsymbol{Q}_i}^\top\|_2 \leq 2 \times \epsilon^{-\frac{1}{2}} \times \|\boldsymbol{W}^Q\|_2$

$\|P_{ii}\|_2 = 1$

$\|\boldsymbol{E}_{ji}\|_2 = 1$

$\|\widetilde{\boldsymbol{V}_j}\|_2 \leq 2 \times \epsilon^{-\frac{1}{2}} \times \|\boldsymbol{W}^{V^\top}\|_2$

Substituting the above results into Eq. 16 and changing $L_\infty$ norm to $L_2$ norm, we have

$$\begin{aligned}
\text{Lip}(SCSA)_2 &= \nu\tau\sqrt{N}\sqrt{N}\frac{N-1}{N}2\epsilon^{-\frac{1}{2}}\left[\|\boldsymbol{W}^K\|_2 + \|\boldsymbol{W}^Q\|_2\right] + 2\nu\epsilon^{-\frac{1}{2}}\|\boldsymbol{W}^{V^\top}\|_2 + \\
&\quad (N-1)\left[\frac{N-1}{N}\nu\tau\sqrt{N}\sqrt{N}2\epsilon^{-\frac{1}{2}}\|\boldsymbol{W}^K\|_2 + 2\nu\epsilon^{-\frac{1}{2}}\|\boldsymbol{W}^{V^\top}\|_2\right] \\
&= 2N(N-1)\nu\tau\epsilon^{-\frac{1}{2}}\|\boldsymbol{W}^K\|_2 + 2(N-1)\nu\tau\epsilon^{-\frac{1}{2}}\|\boldsymbol{W}^Q\|_2 + 2N\nu\epsilon^{-\frac{1}{2}}\|\boldsymbol{W}^{V^\top}\|_2.
\end{aligned}$$

From the upper bound above, we highlight the following observations: 1) $\epsilon$ is to guarantee validity of cosine similarity computation when any participating vector is equal to zero; 2) In $\text{Lip}(SCSA)_2$, the scale factor for the first term is $2N(N-1)\nu\tau\epsilon^{-\frac{1}{2}}$, which multiplies with an extra $\sim N$ when compared to the other terms, meaning that $\|\boldsymbol{W}^K\|_2$ plays a more significant role in the Lipschitz constant of $\text{Lip}(SCSA)_2$.

Different from the L2 distance attention [28], to promise the module is Lipschitz continuous, the scaled cosine similarity attention has *no requirement* for the weight matrices, but the L2 distance attention detailed in [28] requires that $\boldsymbol{W}^Q$ and $\boldsymbol{W}^K$ should be the same.

## H.3 Comparison with Dot-Product Attention [49] and L2-Attention [28]

As proved in [28], the standard dot-product attention is not Lipschitz continuous. The proposed L2-attention is also not Lipschitz continuous for general $\boldsymbol{W}^Q$ and $\boldsymbol{W}^K$, but only Lipschitz continuous when $\boldsymbol{W}^Q = \boldsymbol{W}^K$. However, enforcing $\boldsymbol{W}^Q = \boldsymbol{W}^K$ degrades model performance as shown in [28]. As proved above, the scaled cosine similarity attention (SCSA) is in general Lipschitz continuous, only requiring that $\boldsymbol{W}^Q, \boldsymbol{W}^K, \boldsymbol{W}^V$ have bounded norm and that the computation of cosine similarity is valid. We can easily guarantee that our computation of similarity is valid by introducing a small smoothing factor $\epsilon$.

## I    PROOF OF THEOREM 2

In this section, we give the upper bound on LipsFormer's Lipschitz constant.

For a LipsFormer with $S$ stages where the $s$-th stage has $M_s$ residual blocks, its Lipschitz constant is upper bounded by the inequality below,

$$\mathrm{Lip}(F) \leq \prod_{s=1}^{S} \prod_{m=1}^{M_s} (1 + \alpha_{s,m} \mathrm{Lip}(f_{s,m})). \tag{19}$$

Here, we define $\kappa = \max(\{\mathrm{Lips}(f_i) : i = 1, \ldots, \sum_{s=1}^{S} M_s\})$. When $\alpha$ is set to $\frac{1}{\sum_{s=1}^{S} M_s}$, the above inequality can be rewritten as,

$$\mathrm{Lip}(F) \leq \prod_{s=1}^{S} \prod_{m=1}^{M_s} (1 + \frac{1}{\sum_{s=1}^{S} M_s} \mathrm{Lip}(f_{s,m})) \leq \prod_{s=1}^{S} \prod_{m=1}^{M_s} (1 + \frac{1}{\sum_{s=1}^{S} M_s} \kappa)$$
$$= (1 + \frac{\kappa}{\sum_{s=1}^{S} M_s})^{\sum_{s=1}^{S} M_s} \leq \exp(\kappa). \quad \square \tag{20}$$

## J    DROPPATH IS AN EFFICIENT WAY TO CONSTRAINT THE LIPSCHITZ CONSTANT

DropPath [29] is another effective technique for training deep transformers, where

$$y = \begin{cases} x, & \text{if residual path is dropped} \\ x + \alpha \cdot f(x) & \text{otherwise} \end{cases}$$

When using DropPath with drop probability $p$ within each residual block, the Lipschitz constant of LipsFormer is refined as,

$$\mathrm{Lip}(F) \leq \prod_{s=1}^{S} \prod_{m=1}^{M_s} (1 + \mathrm{droppath}(\alpha_{s,m} \mathrm{Lip}(f_{s,m})), p)), \tag{21}$$

where $\mathrm{droppath}(\alpha_{s,m}, p) = \begin{cases} 0, & \text{with probability } p \\ \alpha_{s,m} \mathrm{Lip}(f_{s,m})) & \text{with probability } 1 - p \end{cases}$.

We can see that DropPath effectively decreases the upper bound of a network's Lipschitz constant by randomly dropping the contributions of residual paths.

## K    COMPARISON WITH EXISTING WORKS

In this section, to clarify our contribution more clearly, we provide a detailed comparison of our method with existing works, including Admin [33], ReZero [3], Swin-V2 [34], DeepNorm [51], L2 self-attention [28] and Spectral Normalization [56].

Admin [33] identifies that within a residual block, the residual branch amplifies network output and the amplification effect makes training unstable. They propose to initialize the weight contributions of a residual branch according to the variance of its previous layer.

ReZero [3] introduces an effective strategy to improve training stability. They notice that initializing the residual branch with 0 satisfies initial dynamical isometry, thus stabilizes model training. ReZero demonstrates that they can train very deep transformer without warmup but it requires removing Layer Normalization. According to Equation 19, initializing the residual contribution to 0 trivially constraints network Lipschitz constant. However, with Layer Normalization back into the network, ReZero is likely to encounter training instability again.

DeepNorm [51] shares similar motivation with Admin [33] and analyzes the influence of the residual block and initialization. They introduce a new normalization function to modify the residual connection in Transformer and propose a new initialization method. However, we observe that learning rate warmup is still necessary in DeepNorm [51].

Training of Admin [33] and DeepNorm [51] requires learning rate warmup, ReZero [3] could train without learning rate warmup but requires that LayerNorm is not present in the network. The analyses of Admin, ReZero, and DeepNorm are not from the perspective of Lipschitz continuity.

In [56], Yuichi *et al.*introduce a simple and effective spectral norm regularization, which penalizes high spectral norm of weight matrices in neural networks. This work focuses on regularization without considering residual block and self-attention block.

In [28], Kim *et al.*prove that the standard dot-product self-attention is not Lipschitz continuous. They introduce an alternative L2 self-attention that is Lipschitz continuous under the constraint that $\boldsymbol{W}^Q = \boldsymbol{W}^K$. Such constraint limits expressiveness of the attention block and empirically degrades training performance. Also, L2 self-attention focuses only on the Lipschitz continuity of self-attention block.

Swin-V2 [34] introduces two strategies to improve training stability of transformer model, including replacing post-norm with pre-norm and a scaled cosine attention replacing the original dot product attention. The introduced scale cosine attention is defined as,

$$\text{Sim}\left(\mathbf{q}_i, \mathbf{k}_j\right) = \cos\left(\mathbf{q}_i, \mathbf{k}_j\right)/\tau + B_{ij}.$$

It should be noted that there's a difference between $\cos\left(\mathbf{q}_i, \mathbf{k}_j\right)/\tau$ and $\tau\cos\left(\mathbf{q}_i, \mathbf{k}_j\right)$, the former is not a Lipschitz continuous function with respect to variable $\tau$ but the latter is. According to our derivation, self-attention based on the scaled cosine attention defined as in Swin-V2 is not Lipschitz continuous if $\boldsymbol{V}$ is not normalized.

The above-mentioned works only deal with one or several standard neural computation modules. Our LipsFormer gives a holistic Lipschitz analysis of a typical transformer network instead of focusing exclusively on a single or few constituent modules. To derive the Lipschitz constant of LipsFormer, we analyze each constituent module of a standard transformer, including convolutions, fully-connected layer, self-attention, normalization and residual block. In this work we propose a Lipschitz continuous self-attention and construct a Lipschitz continuous transformer network by bounding each constituent computation layer. The resultant LipsFormer induces stable training and does not require learning rate warmup.

We summarize our contributions from both theoretical and empirical perspectives as follows, Theoretically,

- We derive a theoretical Lipschitz constant upper bound for scaled cosine similarity attention. Meanwhile, we give a thorough analysis of key Transformer components: LayerNorm, self-attention, residual shortcut, and weight initialization.
- We propose a Lipschitz continuous Transformer (LipsFormer), and derive a theoretical Lipschitz constant upper bound for LipsFormer. The derivation provides a principled guidance for designing LipsFormer networks.

Empirically,

- We make an assumption about the Lipschitz continuity of the network, and experimentally validate this assumption.
- We build LipsFormer on CSwin and Swin-Transformer. We validate the efficacy of the different versions (Tiny, Small, Base, Large and Large++) of LipsFormer on ImageNet, ImageNet-v2 and ImageNet-Real data sets.

