# OpenReview forum: "LipsFormer: Introducing Lipschitz Continuity to Vision Transformers"
_ICLR.cc/2023/Conference — ICLR 2023 poster_

### Official Review · Reviewer_H1uW · 2022-10-24

**Confidence:** 5
**Correctness:** 3
**Technical Novelty And Significance:** 3
**Empirical Novelty And Significance:** 3
**Recommendation:** 6

**Clarity, Quality, Novelty And Reproducibility:**

The paper is well-written. The quality and novelty of the paper are not high enough. The details on the experiment settings are useful, but there is no submitted code to reproduce the results.

**Details Of Ethics Concerns:**

I have no ethics concerns for this paper.

**Strength And Weaknesses:**

**Strong points:**

1. The paper addresses an important issue in designing and training transformers, which is how to guarantee that the model is Lipschitz continuous.

2. The paper provides an interesting Lipschitz constant upper bound for the proposed LipsFormer.

3. The paper is well-written with illustrative figures.

**Weak points:**

1. Experiments on more tasks and different data modalities are needed. Also, results for applying LipsFormer on other baseline vision transformers are missing.

2. There is no result showing that the training is stabilized. The authors should show the training curves in the paper.

3. The authors say, “we show that Lipschitz continuity is a more essential property to ensure training stability,” but there is no comparison to other methods that stabilize the training of vision transformers in the paper. The authors seem to try to provide this comparison in Table 3, but this table is just an ablation study of LipsFormer and does not show that “Lipschitz continuity is a more essential property”.

4. The authors do not empirically compare their method with other related works that introduce Lipschitz continuity to transformers, such as [1]. How is the upper bound of the Lipschitz constant of LipsFormer compared to the bounds from these works?

5. In both Scaled Cosine Similarity Attention and Weighted Residual Shortcut, the predefined or learnable parameters can be absorbed into the weights, W^{Q}, W^{K}, W^{V} or W, to recover the standard dot-product attention and residual connection. Also, the Weighted Residual Shortcut has been explored in [2]. Thus, these two components are not novel.

**Additional Concerns and Questions for the Authors:**

1. Compared to the baseline CSwin Transformer, the LipsFormer has more parameters and more FLOPs. I am not sure if the advantage of LipsFormer over the baseline CSwin Transformer is from the Lipschitz continuity or just because the LipsFormer has a larger size and does more computation. I mention CSwin Transformer here because the authors adopt a similar training strategy as in CSwin Transformer [3].

2. The authors say, “The derivation provides a principled guidance for designing LipsFormer networks (e.g. choosing the weight of a residual shortcut according to the network depth)”. Is there any result to support that the value of the weight of a residual shortcut decided by the network depth is better than other values?

**Minor Comments that did not Impact the Score:**

1. What is 11?

**References:**

[1] Hyunjik Kim, George Papamakarios, and Andriy Mnih. The lipschitz constant of self-attention. In International Conference on Machine Learning, pages 5562–5571. PMLR, 2021.

[2] Falong Shen, Rui Gan, and Gang Zeng. Weighted residuals for very deep networks. In 2016 3rd International Conference on Systems and Informatics (ICSAI), pp. 936-941. IEEE, 2016.

[3] Xiaoyi Dong, Jianmin Bao, Dongdong Chen, Weiming Zhang, Nenghai Yu, Lu Yuan, Dong Chen, and Baining Guo. Cswin transformer: A general vision transformer backbone with cross-shaped windows. arXiv preprint arXiv:2107.00652, 2021.



**Summary Of The Paper:**

This paper proposes the Lipschitz continuous transformer, namely LipsFormer, to stabilize the training of vision transformers. The authors study the key components of a transformer layer, including the LayerNorm, self-attention, residual shortcut, and weight initialization. In LipsFormer, the authors derive the Lipschitz continuous counterparts of the aforementioned components, such as the CenterNorm, scaled cosine similarity function, scaled residual shortcut, and spectral-based initialization. A theoretical Lipschitz constant upper bound for LipsFormer is provided in the paper. The authors justify the advantage of the proposed method on the ImageNet classification task.


**Summary Of The Review:**

Overall, I vote for rejecting. I have two major concerns. First, the proposed Lipschitz continuous components are not novel. Second, there is a significant lack of experimental results to validate the advantage of LipsFormer.

*********** After Discussion ***************

The authors' response largely addresses my concerns. I have increased my original score to 6.

---

> ### Author Response · Authors · 2022-11-18
> **Author response to reviewer H1uW (Part 1)**
>
> Thank you very much for your valuable and constructive suggestions. We have provided more experiments, further clarified our theoretical and empirical contributions and revised the paper according to your comments. We provide detailed responses for each weakness you kindly pointed out, hopefully to address your concerns.
>
> \
> &nbsp;
>
> ### Response to your key concern and our improvements
>
>
> _"You suggest applying LipsFormer on other baseline vision transformer."_
>
> **Our response**: in the revised paper, we have applied our LipsFormer components on Swin-Transformer for a new LipsFormer-Swin network. We evaluate different versions (Tiny, Small, Base and Large) of LipsFormer-Swin training on ImageNet-1K, and show that all models can be trained stablely (thanks to the Lipschitz properties) with competitive results. Meanwhile, we also add more evaluations of our LipsFormer on ImageNet-v2 and ImageNet-Real datasets.
>
> &nbsp;
>
> _"You suggest adding training curves in the paper."_
>
> **Our response**: thank you very much for the suggestion, we have added some related training curves including loss and training accuracy along with the training epoch in Figure 4 in the revised paper.
>
> &nbsp;
>
> _"You suggest adding theoretical and empirical comparison with L2 self-attention."_
>
> **Our response**: thank you very much for the suggestion. We have added experimental comparison with L2 self-attention and clarified our theoretical differences.
>
> **Theoretically.**
> For the L2 self-attention (L2SA), to **guarantee** its Lipschitz continuity, one **necessary condition** is $W_Q = W_K$ as proved in the original L2 self-attention paper.
>
> For the Scaled Cosine Similarity Attention (SCSA),
> $$
> \operatorname{SCSA}(\boldsymbol{X}, \boldsymbol{W}^{Q}, \boldsymbol{W}^{K}, \boldsymbol{W}^{V}, \nu, \tau) =  \nu \boldsymbol{P} \boldsymbol{V},
> \text{where }
> \boldsymbol{P} = \operatorname{softmax}\left(\tau \boldsymbol{Q} \boldsymbol{K}^{\top} \right),
> $$
> where $\nu$ and $\tau$ are predefined or learnable scalars; $\boldsymbol{Q}, \boldsymbol{K}, \boldsymbol{V}$ are $\ell^2$ row-normalized.
>
> Our SCSA **_puts no constraints_** on $W_Q$, $W_K$ and $W_V$. We have clarified this point more clearly in the revised paper.
>
> **Empirically.** We conduct experiments to compare L2 self-attention with our SCSA, as well as the original dot-product attention.
>
>
>
> | Attention  | Acc. |
> | ------------- |:-------------:|
> | Dot Product      | Diverged   |
> | L2 Distance Attn      | 81.3     |
> | SCSA      |81.6   |
>
>
> Besides your suggested experiments, we also add more experiments on large models in the revised paper.

---

> ### Author Response · Authors · 2022-11-18
> **Author response to reviewer H1uW (Part 2)**
>
>
>
>
>
> _"You mention that ``In both Scaled Cosine Similarity Attention and Weighted Residual Shortcut, the predefined or learnable parameters can be absorbed into the weights, $W^{Q}, W^{K}, W^{V}$ or $W$, to recover the standard dot-product attention and residual connection. Also, the Weighted Residual Shortcut has been explored in [2]. Thus, these two components are not novel."_
>
> **Our response**: thank you very much for the comment. We would like to argue that from the perspective of optimization, for both Scaled Cosine Similarity Attention and Weighted Residual Shortcut, the predefined or learnable parameters **cannot** be absorbed into the weights.
>
> For the Weighted Residual Shortcut (WRS),
>
>
> $$
>     \operatorname{WRS}(\boldsymbol{x}, \boldsymbol{W}) = \boldsymbol{x} + \boldsymbol{\alpha} \odot f(\boldsymbol{x}, \boldsymbol{W}),
> $$
> the learnable parameters $\boldsymbol{\alpha}$ cannot be absorbed into a matrix because the current WRS adopts a reparameterization trick that is widely used in deep learning optimization. The optimization problem, when the reparameterization trick is applied, is different from the absorbed version. A classical example of the reparameterization trick (similar to our formulation) is Weight Normalization [1],
> $$\mathbf{w}=g \frac{\mathbf{v}}{\|\mathbf{v}\|},$$
>
> $\text { where } \mathbf{v} \text { is a}  \text { vector, } g \text { is a scalar, and }\|\mathbf{v}\| \text { denotes the Euclidean norm of } \mathbf{v}.$ In the equation, $\mathbf{w}$ is reparameterized into two terms $g$ and $\mathbf{v}$.
> When formulated with two disentangled terms, each term would be individually optimized, inducing a different optimization process.
> One more example of the reparameterization trick  is  as shown in variational Bayesian inference [2]. Therefore, the learnable parameters in Weighted Residual Shortcut cannot be absorbed into the weights.
>
> For the Scaled Cosine Similarity Attention (SCSA),
> $$
> \operatorname{SCSA}(\boldsymbol{X}, \boldsymbol{W}^{Q}, \boldsymbol{W}^{K}, \boldsymbol{W}^{V}, \nu, \tau) =  \nu \boldsymbol{P} \boldsymbol{V},
> \text{where }
> \boldsymbol{P} = \operatorname{softmax}\left(\tau \boldsymbol{Q} \boldsymbol{K}^{\top} \right),
> $$
>
> where $\nu$ and $\tau$ are predefined or learnable scalars; $\boldsymbol{Q}, \boldsymbol{K}, \boldsymbol{V}$ are $\ell^2$ row-normalized: $\mathbf{q}_i, \mathbf{k}_i, \mathbf{v}_i = \frac{{({\mathbf{x}_i}^{\top} \boldsymbol{W}^{Q})}^{\top}}{\sqrt{{\|{\mathbf{x}_i}^{\top} \boldsymbol{W}^{Q} \|}^2 + \epsilon}}, \frac{{({\mathbf{x}_i}^{\top} \boldsymbol{W}^{K})}^{\top}}{\sqrt{{\|{\mathbf{x}_i}^{\top} \boldsymbol{W}^{K} \|}^2 + \epsilon}}, \frac{{({\mathbf{x}_i}^{\top} \boldsymbol{W}^{V})}^{\top}}{\sqrt{{\|{\mathbf{x}_i}^{\top} \boldsymbol{W}^{V} \|}^2 + \epsilon}}$;
>
> the *non-linear transformation* of the cosine similarity in SCSA makes the parameters absorbing not equivalent to our original definition of SCSA. Therefore, the optimization is different between our original SCSA and the suggested absorbed version.

---

> ### Author Response · Authors · 2022-11-18
> **Author response to reviewer H1uW (Part 3)**
>
> ### Response to Additional Concerns and Questions
>
> _"You mention ``Compared to the baseline CSwin Transformer, the LipsFormer has more parameters and more FLOPs. I am not sure if the advantage of LipsFormer over the baseline CSwin Transformer is from the Lipschitz continuity or just because the LipsFormer has a larger size and does more computation."_
>
> **Our response**: The number of parameters for CSwin and LipsFormer-CSwin are 23M and 24M respectively, and the corresponding FLOPs are 4.3G and 4.7G. The increases are arguably negligible.
>
> &nbsp;
>
> _"You ask is there any result to support that the value of the weight of a residual shortcut decided by the network depth is better than other values?"_
>
> **Our response**: Our choice of a residual shortcut weight decided by the network depth is inspired by Lipschitz constraint. Apologies that we do not yet have standalone experiment verifying its effectiveness against other random choices. But we do give a theoretical derivation of the Lipschitz constant of the LipsFormer network based on our Lipschitz continuity assumption of the network and validated the assumption by extensive experiments.
>
>
> \
> &nbsp;
>
> ### Response to Summary Review
> _You mention "I have two major concerns. First, the proposed Lipschitz continuous components are not novel. Second, there is a significant lack of experimental results to validate the advantage of LipsFormer."_
>
> **Our response**: thank you very much for your comments, we have provided more experimental results in the revised paper according to your suggestion. We would like to clarify our contributions more clearly from both theoretical and empirical perspectives.
>
> **Theoretically.**
>
> * We introduce a Lipschitz continuous attention and derive a theoretical Lipschitz constant upper bound for scaled cosine similarity attention. Meanwhile, we give a thorough Lipschitz analysis of key Transformer components: LayerNorm, self-attention, residual shortcut, and weight initialization. We believe these are valuable contributions towards a more stable vision transformer network.
>
> * We propose a Lipschitz continuous Transformer (LipsFormer), and derive a theoretical Lipschitz constant upper bound for LipsFormer. The derivation provides a principled guidance for designing transformer networks that are easier to train (more stable).
>
>
> **Empirically.**
>
> * We make an assumption about the Lipschitz continuity of LipsFormer, and experimentally validated that LipsFormer exhibits good training properties such as more stable training (even \emph{without} warmup) and better classification accuracy.
>
> * We evaluate LipsFormer designs on  Swin-Transformer and CSwin. We validate the efficacy of the the resultant networks of different sizes (Tiny, Small, Base and Large) of LipsFormer-Swin and LipsFormer-CSwin on ImageNet, ImageNet-v2, ImageNet-R datasets showing competitive results.
>
> Hope our clarifications could address your concerns.
>
>
>
> [1] Salimans, Tim, and Durk P. Kingma. "Weight normalization: A simple reparameterization to accelerate training of deep neural networks." Advances in neural information processing systems 29 (2016).
>
> [2] Kingma, Durk P., Tim Salimans, and Max Welling. "Variational dropout and the local reparameterization trick." Advances in neural information processing systems 28 (2015).

---

> > ### Comment · Reviewer_H1uW · 2022-11-19
> > **Reply to the Author’s Rebuttal**
> >
> > Thanks the author for the response. Please find below my concerns and questions after reading the author’s rebuttal.
> >
> > 1. In Table 8, what are the corresponding baselines that the author compares LipsFormer-CSwin-T and LipsFormer-CSwin-B with? Are they Swin-S and Swin-B?
> >
> > 2. In Table 7, the  LipsFormer-Swin models have more parameters and require more FLOPs compared to the baselines but do not improve over the baselines much especially when the model size gets larger. Again, I am not sure if these small advantages are from the Lipschitz Continuity or just because the LipsFormer-Swin models have larger sizes and do more computation.
> >
> > 3. In Figure 4, the authors should also plot the training curves of the baseline Swin models for comparison and to show that the Lipschitz Continuity stabilizes the training.
> >
> > 4. L2 self-attention assumes $W_Q=W_K$. Does it mean L2 self-attention uses fewer parameters while achieving comparable results to the Scaled Cosine Similarity Attention (SCSA)?
> >
> >
> > Since Discussion Stage 1 is almost ended. You can include the answers and results for my question 1 and 2 without the need to put them in the manuscript. I am looking forward to more discussion in Discussion Stage 2.

---

> > > ### Author Response · Authors · 2022-11-21
> > > **Author response to reviewer H1uW**
> > >
> > > We thank the reviewer for the quick reply, which is very helpful for us. We would be happy to address any additional concerns.
> > >
> > >
> > >
> > > **Our response to Q1**: Apologies for not making the purpose of Table 8 more clear. As detailed in Section E, Table 8 is an overfitting evaluation as suggested by reviewer C9zf. As pointed out by previous work [1,2], testing how a method performs in a nearby setting without any finetuning is a good way to assess overfitting. The evaluation is to see how our method, that works well on the original ImageNet validation set, performs on two additional data sets (ImageNet-real and ImageNet-v2). For example,
> > >
> > >
> > > |Mode| Params (M) | Flops (G) | val | real| v2 |
> > > | ------ |:------:|:------:|:------:|:------:|:------:|
> > > | CaiT-B12 | 100.0 | 18.2|  83.3|  87.7|  73.3|
> > > | LipsFormer-CSwin-B | 83|  16.3 | 84.6|  88.6|  74.5|
> > >
> > > According to the above table, the gap between CaiT-B12 and LipsFormer-CSwin-B is 1.3% on ImageNet-val, and the gap on ImageNet-v2 is 1.2\%. It shows that our method does not overfit to the ImageNet validation set.
> > >
> > >
> > > &nbsp;
> > >
> > >
> > > **Our response to Q2**: thank you very much for your question. To address your concern, we would like to provide analyses from two aspects.
> > >
> > > First, we decrease the model size of LipsFormer to be the same as Swin Transformer for performance comparison. For Swin-T, it has [2,2,6,2] blocks in four stages, we use [2,2,5,2] for our LipsFormer-Swin-T to ensure the models have the same number of parameters. The results are shown below,
> > >
> > > |Model | Params (M) | Flops (G) | Blocks | Acc. |
> > > | ------ |:------:|:------:|:------:|:------:|
> > > | Swin-T | 29 | 4.5 | [2,2,6,2]| 81.2|
> > > | LipsFormer-Swin-T | 29|  4.5 | [2,2,5,2]|  82.4 |
> > >
> > > We can see that LipsFormer-Swin-T still shows superior performance.
> > >
> > > Second, we would like to list some relevant works to get a grasp on how the number of the parameters affect model performance.
> > >
> > >
> > > |Model | Params (M) | Flops (G)  | Acc. |
> > > | ------ |:------:|:------:|:------:|
> > > | Swin-T | 29 | 4.5 | 81.2|
> > > |Twins-PCPVT-B [3] | 43.8  |  6.7 | 82.7 |
> > > | LipsFormer-Swin-T | 31|  4.9 | 82.7 |
> > >
> > > It's obvious that LipsFormer improvements bring significant increase in performance with only mild increase in parameters and Flops.
> > >
> > >
> > >
> > > &nbsp;
> > >
> > > **Our response to Q3**: thank you very much for your suggestion. We will include it in the new revision. Meanwhile, we would like to point out the following observations: 1). Swin Transformer cannot be trained **_without learning rate warmup_** due to Nan or Exploding; 2). Our LipsFormer **_can_** be trained stably without learning rate warmup. This observation validates that good Lipschitz continuity of a network can make the training stabilized without training stabilization tricks such as learning rate warmup.
> > >
> > > &nbsp;
> > >
> > > **Our response to Q4**: This is a very good question and the short answer is no. In fact, L2 distance attention uses the same number of parameters as SCSA, including weight matrices $W^Q, W^K, W^V$ and a temperature $\tau$ (a scalar).
> > >
> > > As proved in the paper [4], to ensure Lipschitz continuity of L2 distance attention, it is required that $W^Q = W^K$. However, enforcing $W^Q = W^K$ limits the expressiveness of the L2 distance attention resulting in inferior performance, extensive ablation study was conducted in the paper for this point. Therefore, in practice, they do not rigorously enforce the equality of $W^Q$ and $W^K$.
> > >
> > >
> > >
> > >
> > > [1] Touvron, Hugo, Andrea Vedaldi, Matthijs Douze, and Hervé Jégou. "Fixing the train-test resolution discrepancy." Advances in Neural Information Processing Systems 32 (2019).
> > >
> > > [2] Touvron, Hugo, Matthieu Cord, and Hervé Jégou. "Deit iii: Revenge of the vit." arXiv preprint arXiv:2204.07118 (2022).
> > >
> > > [3] Chu, Xiangxiang, Zhi Tian, Yuqing Wang, Bo Zhang, Haibing Ren, Xiaolin Wei, Huaxia Xia, and Chunhua Shen. "Twins: Revisiting the design of spatial attention in vision transformers." Advances in Neural Information Processing Systems 34 (2021): 9355-9366.
> > >
> > > [4] Kim, Hyunjik, George Papamakarios, and Andriy Mnih. "The lipschitz constant of self-attention." In International Conference on Machine Learning, pp. 5562-5571. PMLR, 2021.

---

> > > > ### Comment · Reviewer_H1uW · 2022-12-04
> > > > **Thanks for the response!**
> > > >
> > > > The authors' response largely addresses my concerns. I have increased my original score to 6.

---

### Official Review · Reviewer_HiVm · 2022-10-24

**Confidence:** 3
**Correctness:** 3
**Technical Novelty And Significance:** 4
**Empirical Novelty And Significance:** 4
**Recommendation:** 8

**Clarity, Quality, Novelty And Reproducibility:**

This paper is mostly well written. The proposed techniques are novel and seem to perform well empirically.
When presenting the definition of each operations, it'd be better to also give the dimension of the input, output and weight matrices. For instance, the definition of LayerNormalization is for $x \in \mathbb{R}^D$, and the equation will look different when the input is a high dimensional tensor.

**Strength And Weaknesses:**

Strength:

This paper analyzes the Lipschitz bound of each component module of ViT comprehensively, and proposed Lipschitz counterparts for the non-Lipschitz modules.
Extensive show superior performance compared with previous works, and ablation studies show the importance of each proposed module.

Weaknesses:
1) This paper replies on the assumption that training stability relies on Lipschitz continuity. My main question is: whether introducing Lipschitz continuity will cause other issues for training stability, since training stability also depends on other aspects other than Lipschitz continuity. For instance, for instance, BatchNorm and LayerNorm are proposed to relieve the "covariate shift" problem. The proposed CenterNorm throw out std(y) in the denominator, will this cause other issues due to "covariate shift" of neurons from different layers? It may be good to check how does std(y) change across layers with different normalizations.

2) How does scaled cosine similarity perform compared with L2 self-attention proposed in Kim 2021, given these two operations are both Lipschitz continuous? Either theoretical or empirical justification would be fine.

3) Since some of the proposed modules are initialization techniques, I think it'd be interesting to see how the learnable parameters change during training. For instance, does the weight of skipping connection $\alpha$ stays on a similar order of magnitude as one initializes it?

**Summary Of The Paper:**

This paper proposed a new ViT architecture called LipsFormer, which replaces non-Lipschitz or unstable modules with Lipschitz continuous counterparts. Experiments show that LipsFormer obtains better Top-1 accuracy on Imagenet-1k dataset compared with previous works.

**Summary Of The Review:**

This paper proposed LipsFormer, which enforces Lipschitz continuity on the submodules of ViT. The proposed techniques show good empirical performances and seem to be effective In stabilizing the training of ViT.

---

> ### Author Response · Authors · 2022-11-18
> **Author response to reviewer HiVm**
>
>
>
>
> Thank you very much for your encouraging review and constructive suggestions. We have added more experiments and revised the paper according to your comments. We would like to provide  detailed responses for your questions.
>
> &nbsp;
>
> ### Response to your key concern and our improvements
>
>
> _"You ask whether the proposed CenterNorm that removes std(y) in the denominator will cause other issues due to ``covariate shift'' of neurons from different layers?"_
>
> **Our response**:  Shibani et al. provided an intriguing perspective in their paper [1] that BatchNorm helps optimization not because it relieves ``covariate shift'' problem, but that _"We demonstrate that such distributional stability of layer inputs has little to do with the success of BatchNorm. Instead, we uncover a more fundamental impact of BatchNorm on the training process: it makes the optimization landscape significantly smoother."_ Lipschitz continuity with bounded Lipschitz constant also aims to encourage a smoother function and subsequently a smoother optimization landscape.
>
> In vision Transformers, given an input feature map y of shape $B\times H\times W\times C$, the statistics of Std(y) for each channel in BatchNorm is usually conducted on $B\times H \times W$, while the statistics of Std(y) for LayerNorm is usually conducted on $C$ only. Since $C$ is of much lower dimension than  $B\times H\times W$, Std(y) for LayerNorm is more prone to abnormal (high variance) std value, leading to a non-smooth landscape with high probability. Our proposed CenterNorm is a smoothed version of Normalization that has bounded Lipschitz constant.
>
>
> &nbsp;
>
> _"You suggest adding  comparison with the L2 self-attention, either theoretical and empirical comparison would be fine."_
>
> **Our response**: thank you very much for the suggestion. We have added experimental comparison with L2 self-attention and clarified our theoretical differences.
>
> **Theoretically.**
> For the L2 self-attention (L2SA), to **guarantee** its Lipschitz continuity, one **necessary condition** is $W_Q = W_K$ as proved in the original L2 self-attention paper.
>
> For the Scaled Cosine Similarity Attention (SCSA),
> $$
> \operatorname{SCSA}(\boldsymbol{X}, \boldsymbol{W}^{Q}, \boldsymbol{W}^{K}, \boldsymbol{W}^{V}, \nu, \tau) =  \nu \boldsymbol{P} \boldsymbol{V},
> \text{where }
> \boldsymbol{P} = \operatorname{softmax}\left(\tau \boldsymbol{Q} \boldsymbol{K}^{\top} \right),
> $$
> where $\nu$ and $\tau$ are predefined or learnable scalars; $\boldsymbol{Q}, \boldsymbol{K}, \boldsymbol{V}$ are $\ell^2$ row-normalized.
>
> Our SCSA **_puts no constraints_** on $W_Q$, $W_K$ and $W_V$. We have clarified this point more clearly in the revised paper.
>
> **Empirically.** We conduct experiments to compare L2 self-attention with our SCSA, as well as the original dot-product attention.
>
>
>
> | Attention  | Acc. |
> | ------------- |:-------------:|
> | Dot Product      | Diverged   |
> | L2 Distance Attn      | 81.3     |
> | SCSA      |81.6   |
>
>
> Besides your suggested experiments, we also add more experiments on large models in the revised paper.
>
> &nbsp;
>
> _"You suggest providing how learnable parameters (e.g., $\boldsymbol{\alpha}$) change during training."_
>
> **Our response**: thank you for your helpful suggestion. We now add a figure to show how $\boldsymbol{\alpha}$ changes during training. Meanwhile, we also provide the training curve of our model for more clarity on the training process.
>
>
> \
> &nbsp;
>
> ### Response to Clarity, Quality, Novelty And Reproducibility
>
> _"You suggest ``It would be better to also give the dimension of the input, output and weight matrices. For instance, the definition of Layer Normalization is for $\mathbf{x} \in \mathbb{R}^{D}$, and the equation will look different when the input is a high dimensional tensor."_
>
> **Our response**: we have revised the paper according to your comments.
>
> \
> &nbsp;
>
> ### Response to Summary Review
> We have supplied more experimental results and comparisons. Thank you for all the helpful suggestions.
>
> &nbsp;
>
> [1] Santurkar, Shibani, Dimitris Tsipras, Andrew Ilyas, and Aleksander Madry. "How does batch normalization help optimization?." Advances in Neural Information Processing Systems 31 (2018).

---

### Official Review · Reviewer_C9zf · 2022-10-25

**Confidence:** 4
**Correctness:** 3
**Technical Novelty And Significance:** 4
**Empirical Novelty And Significance:** 3
**Recommendation:** 6

**Clarity, Quality, Novelty And Reproducibility:**

I am satisfied with the quality, and also like this novel and interesting idea of introducing Lipschitz continuous into vision transformer.

**Strength And Weaknesses:**

Strengthen:

++ Writtings are clear in general.

++ The main idea of introducing Lipschitz continuous into vision transformer is novel and interesting.

++ The theoretical analysis indeed strengthens this work.

Weaknesses:

-- Only the performances on ImageNet 1K are reported. It is necessary to report the performances on more downstream tasks like object detection or semantic segmentation.

-- Several recent works of vision transformer are missing in Table 2 for performance comparison:

[A] Crossvit: Cross-attention multi-scale vision transformer for image classification, ICCV. 2021.
[B] Contextual transformer networks for visual recognition, IEEE TPAMI, 2022.
[C] Towards robust vision transformer, CVPR. 2022.
[D] Regionvit: Regional-to-local attention for vision transformers, ICLR. 2022
[E] Wave-ViT: Unifying Wavelet and Transformers for Visual Representation Learning. ECCV, 2022.

-- Only Params and FLOPs are reported. However, it is important to add more metrics of computation cost (like memory consumption and latency)  to better understand the trade-offs.

-- Following DeiT, it is necessary to perform evaluation on ImageNet-v2 [F] to measure the level of overfitting.

[F] Do ImageNet Classifiers Generalize to ImageNet? ICML, 2019.


**Summary Of The Paper:**

This paper presents a Lipschitz continuous Transformer (LipsFormer), that pursues training stability both theoretically and empirically for vision transformer. LipsFormer replaces unstable Transformer component modules with a series of Lipschitz continuous counterparts. Experiments are performed to validate the proposals.

**Summary Of The Review:**

In general, I lean to positive and like this new idea. However, I have several concerns about the insufficient experiments. I hope this issue can be addressed in the next round.

---

> ### Author Response · Authors · 2022-11-18
> **Author response to reviewer C9zf**
>
>
> We thank the reviewer for the helpful suggestions (and happy to see that you like this new idea). We have revised our paper according to your comments.
>
> &nbsp;
>
> ### Response to your key concerns and our improvements
>
>
> _"You suggest reporting the performance on more downstream tasks like object detection or semantic segmentation."_
>
> **Our response**: in the revised paper, we apply our LipsFormer components on Swin-Transformer (named LipsFormer-Swin). We evaluate different versions (Tiny, Small, Base, Large) of LipsFormer-Swin training on ImageNet-1K, and show that all models can be trained smoothly _without_ any warmup. Meanwhile, we add more evaluations of our LipsFormer on ImageNet-v2 and ImageNet-Real datasets. Apologies for not finishing experiments on more new tasks due to the limited time. We plan to include more downstream tasks in the future.
>
> &nbsp;
>
> _"You suggest adding more references for comparison."_
>
> **Our response**: we now include your suggested works for comparison in the revised version. Thank you for your valuable comments.
>
> &nbsp;
>
> _"You suggest adding more metrics of computational cost to better understand the trade-offs."_
>
> **Our response**: we now report our training costs, including training FPS and memory consumption in the revised paper.
>
> &nbsp;
>
> _"You suggest performing evaluation on more data sets."_
>
> **Our response**: thank you for the valuable suggestions. We now add evaluations on two new datasets, including ImageNet-v2 and ImageNet-Real. The results are shown below.
>
>
> |Mode| Params (M) | Flops (G) | val | real| v2 |
> | ------ |:------:|:------:|:------:|:------:|:------:|
> | ViT-S | 22.0 | 4.6  | 80.4 | 86.1 | 69.7 |
> | PiT-S | 23.5 | 2.9 | 80.4 | 86.1 | 69.2|
> | TNT-S | 23.8 | 5.2 | 81.4 | 87.2 | 70.6|
> | ConViT-S | 27.8 | 5.8 | 81.3 | 87.0 | 70.3|
> | Swin-S  | 49.6 | 8.7 | 82.1 | 86.9 | 70.7|
> | LipsFormer-CSwin-T |  24 | 4.7 | 83.5 | 88.0|  73.2|
> | ViT-B | 86.6 | 17.6|  83.1 | 87.7|  72.6|
> | PiT-B | 73.8|  12.5|  82.4|  86.8|  72.0|
> | TNT-B | 65.6|  14.1|  82.9|  87.6 | 72.2|
> | ConViT-B | 86.5 | 17.5 | 82.0 | 86.7|  71.3|
> | Swin-B | 87.8|  15.4 | 82.2 | 86.7 | 70.7|
> | CaiT-B12 | 100.0 | 18.2|  83.3|  87.7|  73.3|
> | LipsFormer-CSwin-B | 83|  16.3 | 84.6|  88.6|  74.5|
>
> \
> &nbsp;
>
> ### Response to Summary Review
> _You mention "In general, I lean to positive and like this new idea. However, I have several concerns about the insufficient experiments. I hope this issue can be addressed in the next round."_
>
> **Our response**: We now include more experiments according to your suggestion:
> * We apply our LipsFormer components on Swin-Transformer for a LipsFormer-Swin network. We evaluate different versions (Tiny, Small, Base and Large) of LipsFormer-Swin training on ImageNet-1K, and show that all models can be trained stablely with competitive results.
> * We add more evaluations of our LipsFormer on ImageNet-v2 and ImageNet-Real datasets. Apologies for not finishing experiments on more new tasks due to the limited time. We plan to include more downstream tasks in the future.
>
>
> Hope the new experimental results would address your concerns.

---

### Official Review · Reviewer_CmFG · 2022-10-26

**Confidence:** 2
**Correctness:** 3
**Technical Novelty And Significance:** 2
**Empirical Novelty And Significance:** 3
**Recommendation:** 6

**Clarity, Quality, Novelty And Reproducibility:**

Clarity: The paper is well-written and I enjoy reading it.

Quality: The paper contains well-motivated modules and verifies the efficiency through experiments.

Novelty: Although several models are already considered in several previous works (e.g., ReZero, Swin-V2, and DeepNorm), this paper combines them together and simplifies the structure with theoretical analysis. The better numerical performance verifies that those modules are effective.

Reproducibility:  The training details are provided in Appendix C but the code is not provided.

**Strength And Weaknesses:**

Strength:
Simple but effective structures are proposed which are also well-motivated theoretically.

Weaknesses:
In the numerical tests, the model size considered is relatively small (e.g., < 100 M). Usually, in highly overparameterization regimes (e.g., > 300M) the landscape or geometry in terms of optimization is quite different from those in the small model regimes. Many training stabling/accelerating tricks that work well with small models may fail to be effective. It is worth evaluating whether the proposed modules can also improve the training for the large enough model (e.g., >300M) and push a higher SOTA accuracy.


**Summary Of The Paper:**

This paper introduces several modules to stabilize and accelerate the training of the vision transformers. In particular, the CenterNorm, scaled cosine similarity attention, and spectral initialization are proposed, which are the counterpart of the LayerNorm, standard self-attention, and  Xavier/Kaiming initialization respectively. The underlying philosophy is that model's Lipschitz constant plays a critical role in stabilizing the training procedure. The smaller Lipschitz constant enables us to the deeper networks and it is essential to design the structure/modules to control the changes in the Lipschitz constant. The authors verify the efficiency of the proposed modules in the ImageNet-1K classification task and in the ablation study the effect of each individual module is discussed.

**Summary Of The Review:**

This paper provides several modules to control the Lipschitz constant of the Transformer models. The modules are theoretically justified and show promising numerical performance in ImageNet-K classification. It would be better if we could see the experiments on the large model (e.g., >300M parameters).

Minor issue:

1. Please add the definition to the $\odot$ operator in sec. 4.1.1.
2. Please add a reference for Droppath in the main paper.


Based on the current presentation, I tend to accept it and I'm willing to change my evaluation after the rebuttal.

---

> ### Author Response · Authors · 2022-11-18
> **Author response to reviewer CmFG**
>
> We thank the reviewer for the constructive suggestions (and happy to know that you enjoy reading our work). We have added more experimental results according to your comments.
>
> \
> &nbsp;
>
>
> ### Response to your key concerns and our improvements
> _"Your main concern is whether the LipsFormer can stablely generalize to large model (e.g., >300M)"_
>
> **Our response**: in the rebuttal stage, we apply LipsFormer components on Swin Transformer (named LipsFormer-Swin), and conduct extensive experiments on LipsFormer-Swin of different model sizes from Tiny (31 M), to Small (52M), Base (96M), Large (214M) and Large++ (526M) training on ImageNet-1K. We summarize our findings as below.
>
> * All LipsFormer-Swin models, **_without_** any warmup, can stablely and successfully converge with competitive performance. We provide training curve of the largest model (LipsFormer-Swin Large++) in the revised paper.
>
>
> * Large model needs large scale data to avoid overfitting.
> We train our Tiny, Small, Base, Large and Large++ models on ImageNet-1K and already observe that LipsFormer-Swin-Large shows obvious overfitting (the training loss is smaller, but the test accuracy is lower) on ImageNet-1K. Similarly as discussed in github
> [github issue](https://github.com/microsoft/Swin-Transformer/issues/261), it has also been observed that the original Swin-L cannot outperform Swin-B when training ONLY on ImageNet-1K. In our experiment, the LipsFormer-Swin Large++ can be smoothly trained on ImageNet-1K _without_ any learning rate warmup in spite of some overfitting.
>
> \
> &nbsp;
>
>
>
> ### Response to Clarity, Quality, Novelty and Reproducibility
>
> **Our response**: thank your very much for your encouraging comments. We plan to release our code after it is accepted for publication.
>
> \
> &nbsp;
>
> ### Response to Summary Review
> _You mention ``This paper provides several modules to control the Lipschitz constant of the Transformer models. The modules are theoretically justified and show promising numerical performance in ImageNet-K classification. It would be better if we could see the experiments on the large model (e.g., > 300M parameters).''}_
>
> **Our response**: we train a LipsFormer-Swin-Large++ (526M) model on ImageNet-1K _without_ any learning rate warmup. The training process is smooth in spite of some overfitting. We hope this will address your concerns.
>
>
> \
> &nbsp;
>
> ### About Minor Issue
>
> **Our response**: Thanks for pointing these out. We now add definition of the $\odot$ operator and a reference to Droppath in the main paper.

---

### Author Response · Authors · 2022-11-18
**Summary of Paper Revision**

We would like to thank all the reviewers for their valuable and constructive comments.

To address reviewers' comments and concerns, we have made the following changes:

* We clarify our theoretical and empirical contributions more clearly with newly added experiments in Sec. K.

* We add more experimental results of LipsFormer on ImageNet-v2, ImageNet-R and ImageNet-A in Sec. E.

* We further apply LipsFormer components on Swin-Transformer, which we call LipsFormer-Swin. We conduct extensive experiments on LipsFormer-Swin with varying model sizes: from Tiny (~ 30M) to Large++ (> 500M) in Sec. D.

* We add figures showing how the learnable parameters change during training, as well as training curves in Sec. F and Sec. G.

* We add an experimental comparison between the scaled cosine similarity attention and L2 self-attention. We also further clarify the theoretical difference between them in Sec. H.

We mark the revised content in blue color for your reference.

---

### Decision · Program_Chairs · 2023-01-20

**Decision:**

Accept: poster

**Justification For Why Not Higher Score:**

The reviewers had a variety of misgivings but overall are satisfied.

**Justification For Why Not Lower Score:**

Overall, reviewers are satisfied.

**Metareview: Summary, Strengths And Weaknesses:**

This paper provides a Lipschitz alternative to attention layers, and provides a thorough experimental study.  Overall the reviewers were satisfied with the results and I am also happy to recommend acceptance.

Technical note: I found it a little unsatisfying that mathematical Lipschitz calculations (e.g., Lemma 2) were only presented as upper bounds.  It is fairly easy to construct specific instances where there is a nearly matching lower bound, and I'm uncertain why the authors chose not to do so, as it would have not only strengthened the theory, but also provided a few more refined areas of empirical study; e.g., finding lower bound instances that match the observed lipschitz constants could prove quite tricky.  Another reason to identify lower bounds is that these are particularly easy to find in practice; e.g., with any two explicit points returned by any standard method, one can construct a lower bound, whereas in practice it is tight upper bounds which are much more expensive to compute.

**Note From Pc:**

if the above contains the word "oral" or "spotlight" please see: "oral" presentation means -> notable-top-5% and "spotlight" means -> notable-top-25%. As stated in our emails, we are disassociating presentation type from AC recommendations

---

> ### Public Comment · ~Yi_Rao2 · 2023-02-20
> **Is LayerNorm Lipschitz continuous?**
>
> Thank you for your work, but the claim in your paper that "LayerNorm is not Lipschitz continuous" is inconsistent with the conclusion in paper: The lipschitz constant of self-attention (In Appendix N of this paper, and this paper is also cited as [28] in your paper). I could be wrong, but from the proof it looks like they're right, that LayerNorm is Lipschitz continuous. And I found out experimentally that LayerNorm is Lipschitz continuous. So your conclusion that "LayerNorm is not Lipschitz continuous" may be wrong, which may make your method 4.1.1 CenterNorm meaningless.
> Could you further point out why you proof is correct? Or which of the proof steps in paper [28] went wrong that led to the opposite conclusion?

---

> > ### Public Comment · ~Jiaren_Zhao1 · 2023-05-24
> > **Agree and suspect this paper.**
> >
> > LayerNorm is Lipschitz continuous and it is proven in Kim's L2 Attention paper. We suspect the experiment of LipsFormer is fake and are trying to implement their code and post to somewhere. However, the reviewer of this paper seems not familiar with theory and only discusses some useless aspects. Obviously this paper should not be accepted.

---

> > > ### Author Response · Authors · 2023-05-24
> > > **Please read the definitions carefully.**
> > >
> > > Please read the original LayerNorm (Jimmy Lei Ba et al.) [1] and Kim's paper [2], and the definition in our paper [3] carefully.
> > >
> > > The original LayerNorm definition is,
> > > \begin{equation}
> > > \begin{aligned} L N(\mathbf{z} ; \boldsymbol{\alpha}, \boldsymbol{\beta}) & =\frac{(\mathbf{z}-\mu)}{\sigma} \odot \boldsymbol{\alpha}+\boldsymbol{\beta} \\ \text{where}\ \ \mu=\frac{1}{D} \sum_{i=1}^{D} z_{i}, \quad \sigma =\sqrt{\frac{1}{D} \sum_{i=1}^{D}\left(z_{i}-\mu\right)^{2}}\end{aligned}
> > > \end{equation}
> > >
> > > We use the original definition as [1], we write it as,
> > > \begin{equation*}
> > > \begin{aligned}
> > > \operatorname{LN}(\boldsymbol{x} ) &= \boldsymbol{\gamma} \odot \boldsymbol{z} + \boldsymbol{\beta},
> > > \ \text{where}\ \ \boldsymbol{z} = \frac{\boldsymbol{y} }{\mathrm{Std}(\boldsymbol{y} )} \ \ \text{and}\ \  \boldsymbol{y} = \left(\boldsymbol{I}-\frac{1}{D}  \boldsymbol{1} \boldsymbol{1}^{\top}\right) \boldsymbol{x} ,
> > > \end{aligned}
> > > \end{equation*}
> > >
> > > You can derive its gradient, if the std(y) = 0,  the Lipschitz constant will be $\frac{1}{0.0} = \infty$, it is not Lipschitz continuous.
> > >
> > > In kim's paper, they use a smoothed definition of LayerNorm, it is defined as,
> > > \begin{equation}
> > > \mathrm{LN}(\boldsymbol{x})=\boldsymbol{\gamma} \odot \boldsymbol{z}+\boldsymbol{\beta}, \text{where   } \boldsymbol{z}=\frac{\boldsymbol{y}}{\sqrt{\|\boldsymbol{y}\|_{2}^{2}+\epsilon}} \text{and} \boldsymbol{y}=\left(\boldsymbol{I}-\frac{1}{D} \mathbf{1 1}^{\top}\right) \boldsymbol{x}
> > > \end{equation}
> > >
> > >
> > >
> > > The Jacobian matrix of the variable $\boldsymbol{z}$ with respect to $\boldsymbol{x}$ can be calculated as,
> > >
> > > \begin{equation}
> > > \boldsymbol{J}_{\boldsymbol{z}}(\boldsymbol{x})=\frac{\partial \boldsymbol{z}}{\partial \boldsymbol{x}}
> > > =\frac{1}{\sqrt{ {\|\boldsymbol{y}\|}_2^2 +\epsilon}}  \left(\boldsymbol{I}-\frac{1}{d} \boldsymbol{1} \boldsymbol{1}^{\top}\right)  \left( \boldsymbol{I} - \frac{\boldsymbol{y} \boldsymbol{y}^{\top}}{{\|\boldsymbol{y}\|}_2^2 +\epsilon}   \right)
> > > \end{equation}
> > > its Lipschitz constant under $L_2$ norm is $\(\epsilon^{-\frac{1}{2}} \max _{d}\left(\frac{D^{2}-2}{D}\right)\)$, where $\epsilon$ is very small, such as e-8, it is very large, $\frac{1}{{\|\epsilon\|}^{0.5}}$ is 1e4, D can be large in large model, such as 8192. 8.192*e7, in mixed-precision training, you cannot even represent it (the maximum value can be denoted is 65504 in FP16).  But I still agree that the smoothed version of LN is Lipschitz continuous.
> > >
> > >
> > > Please read the above definitions carefully. Kim's paper is a very good paper, our derivation of scaled cosine attention in Lipsformer is motivated by their excellent work.  I enjoy reading his work, but I am not stupid to challenge one simple and correct concept in his paper. We are considering two different versions of LN, smoothed and non-smoothed.
> > > In Optimization, changing one parameter in the equation leads to different convergence and properties. That will be a different story. Which story do you like?
> > >
> > >
> > > Replied by Xianbiao Qi.
> > >
> > >
> > > [1] Ba, Jimmy Lei, Jamie Ryan Kiros, and Geoffrey E. Hinton. "Layer normalization." arXiv preprint arXiv:1607.06450 (2016).
> > >
> > > [2] Kim, Hyunjik, George Papamakarios, and Andriy Mnih. "The lipschitz constant of self-attention." International Conference on Machine Learning. PMLR, 2021.
> > >
> > > [3] Qi, Xianbiao, et al. "LipsFormer: Introducing Lipschitz Continuity to Vision Transformers." arXiv preprint arXiv:2304.09856 (2023).